# MMBench-GUI: Hierarchical Multi-Platform Evaluation Framework for GUI Agents

## Abstract

We introduce MMBench-GUI, a hierarchical benchmark for evaluating GUI automation agents across Windows, macOS, Linux, iOS, Android, and Web. The benchmark spans four levels: Content Understanding, Element Grounding, Task Automation, and Task Collaboration, covering essential skills for GUI agents. To assess both effectiveness and efficiency, we further propose the Efficiency–Quality-Aware (EQA) metric, which measures task success alongside action redundancy. Extensive evaluations reveal that precise visual grounding is the critical determinant of performance, underscoring the advantages of modular designs with specialized grounding modules. Moreover, all agents suffer from substantial inefficiencies, frequently completing tasks with excessive steps despite eventual success. Performance also degrades on complex or cross-application tasks, exposing weaknesses in memory, planning, and adaptive reasoning. By providing broad coverage, standardized protocols, and novel metrics, MMBench-GUI establishes the first comprehensive foundation for advancing GUI agent research.

## 1 Introduction

The rapid progress of Vision-Language Models (VLMs) (Wang et al., 2024; Chen et al., 2024b; Bai et al., 2025; Zhu et al., 2025; Team et al., 2025; Xiaomi, 2025; Wang et al., 2025b) has greatly advanced GUI agents, enabling complex interactions within graphical interfaces (Wu et al., 2024a; Cheng et al., 2024; Hong et al., 2024; Zheng et al., 2024; Gou et al., 2024). These agents show a strong potential to automate repetitive tasks in different domains, thus improving productivity (Xu et al., 2024b; Wu et al., 2024b; Lin et al., 2024; Qin et al., 2025; Yang et al., 2024).

Nevertheless, existing benchmarks (Zhou et al., 2023; Cheng et al., 2024; Xie et al., 2024; Li et al., 2024; Chang et al., 2024; Rawles et al., 2024; Li et al., 2025; Nayak et al., 2025; Sun et al., 2025; Xie et al., 2025) suffer from three critical limitations: (1) they focus on isolated skills and overlook the relationships among multiple capabilities (Deng et al., 2023a; Cheng et al., 2024; Xie et al., 2025; Li et al., 2025); (2) their metrics prioritize the success rate while neglecting efficiency (Zhou et al., 2023; Xu et al., 2024a; Xie et al., 2024; Bonatti et al., 2024); and (3) their scenario coverage is narrow, failing to represent widely used GUI systems (He et al., 2024; Xie et al., 2024; Rawles et al., 2024; Sun et al., 2025).

To overcome these gaps, we propose MMBench-GUI, a hierarchical, multi-platform benchmark for systematic evaluation of GUI agents. As illustrated in Figure 1, the framework consists of four ascending levels: (1) GUI Content Understanding, (2) GUI Element Grounding, (3) GUI Task Automation, and (4) GUI Task Collaboration. Each level captures essential skills, from basic perception to cross-application collaboration, enabling comprehensive capability assessment. In addition, we introduce the Efficiency–Quality-Aware (EQA) metric, which jointly evaluates accuracy and efficiency by measuring both task success and running steps. Finally, to ensure practical relevance, we construct a dataset spanning six major platforms, thereby reflecting diverse real-world scenarios.

Leveraging extensive evaluations with MMBench-GUI, we identify key limitations in current GUI agents. First, while general language models excel at high-level reasoning and planning, they perform poorly in precise visual interaction. Accurate visual grounding emerges as the primary determinant of task success, underscoring the need for improved localization. Second, efficiency has become a critical bottleneck beyond raw success rate. Our proposed EQA metric highlights the redundant steps prevalent in contemporary agents, caused by localization errors, incomplete action spaces, and

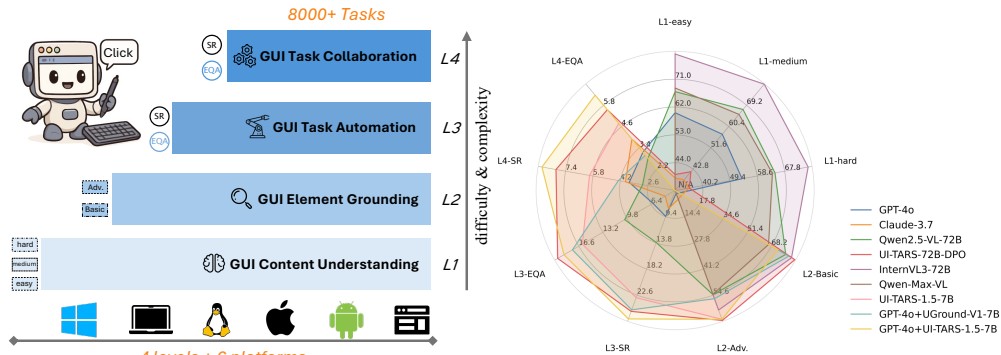

Figure 1: **MMBench-GUI**: a hierarchical benchmark spanning four levels of increasing difficulty, covering over 8,000 tasks across six commonly used platforms. From L1 to L4, task complexity increases progressively, placing growing demands on the agent's generalization and reasoning abilities. Based on this benchmark, we visualize the performance of various models in the right figure, clearly illustrating their respective strengths as well as areas with substantial room for improvement.

shortsighted planning. Third, agent performance degrades substantially in complex, ambiguous, and cross-application tasks, revealing deficiencies in memory, state tracking, and adaptive reasoning. Addressing these shortcomings is essential for advancing GUI agent capabilities.

In summary, our primary contributions are as follows:

- Hierarchical, cross-platform benchmark. We introduce a human-centered, progressive benchmark that evaluates GUI agents across four essential capability levels. For static tasks (L1, L2), we provide fine-grained difficulty stratification; for dynamic tasks (L3, L4), we offer cleaned data splits and novel task constructions to better mirror real-world variability.

- Comprehensive platform coverage. Our benchmark spans all major operating systems: Windows, Linux, macOS, Android, iOS, and the Web, enabling consistent multi-platform evaluation under a unified protocol. To our knowledge, this is the first benchmark to incorporate online task scenarios for macOS, filling a long-standing gap in GUI agent evaluation.

- Novel efficiency-aware metric. We propose the EQA metric to jointly evaluate success and efficiency in online tasks. Unlike prior works that focus solely on success rate, EQA additionally measures whether tasks are completed within a step budget, thereby quantifying action redundancy. This provides deeper insight into agent behavior and encourages the development of agents that are not only capable but also efficient.

## 2 MMBENCH-GUI

### 2.1 HIERARCHICAL EVALUATION

Existing GUI agents typically execute tasks by emulating human operations such as mouse clicks and keyboard inputs, which requires both interface comprehension and long-horizon planning. However, current benchmarks evaluate only isolated aspects—for example, Screenspot (Cheng et al., 2024) emphasizes spatial localization, while OSWorld (Xie et al., 2024) focuses on end-task success—without systematically assessing the full spectrum of underlying competencies. Consequently, the relationships among these abilities remain unclear, making it difficult to disentangle the specific factors that drive agent success or failure.

To overcome these limitations, we propose MMBench-GUI, a hierarchical evaluation framework as shown in Figure 1. The framework comprises four ascending levels: (1) L1-GUI Content Understanding, (2) L2-GUI Element Grounding, (3) L3-GUI Task Automation, and (4) L4-GUI Task Collaboration. Each level introduces tasks of increasing complexity, enabling systematic assessment of agents under progressively more demanding scenarios. The complete benchmark covers over 8,000 tasks across various platforms, with detailed statistics reported in Appendix A.3.

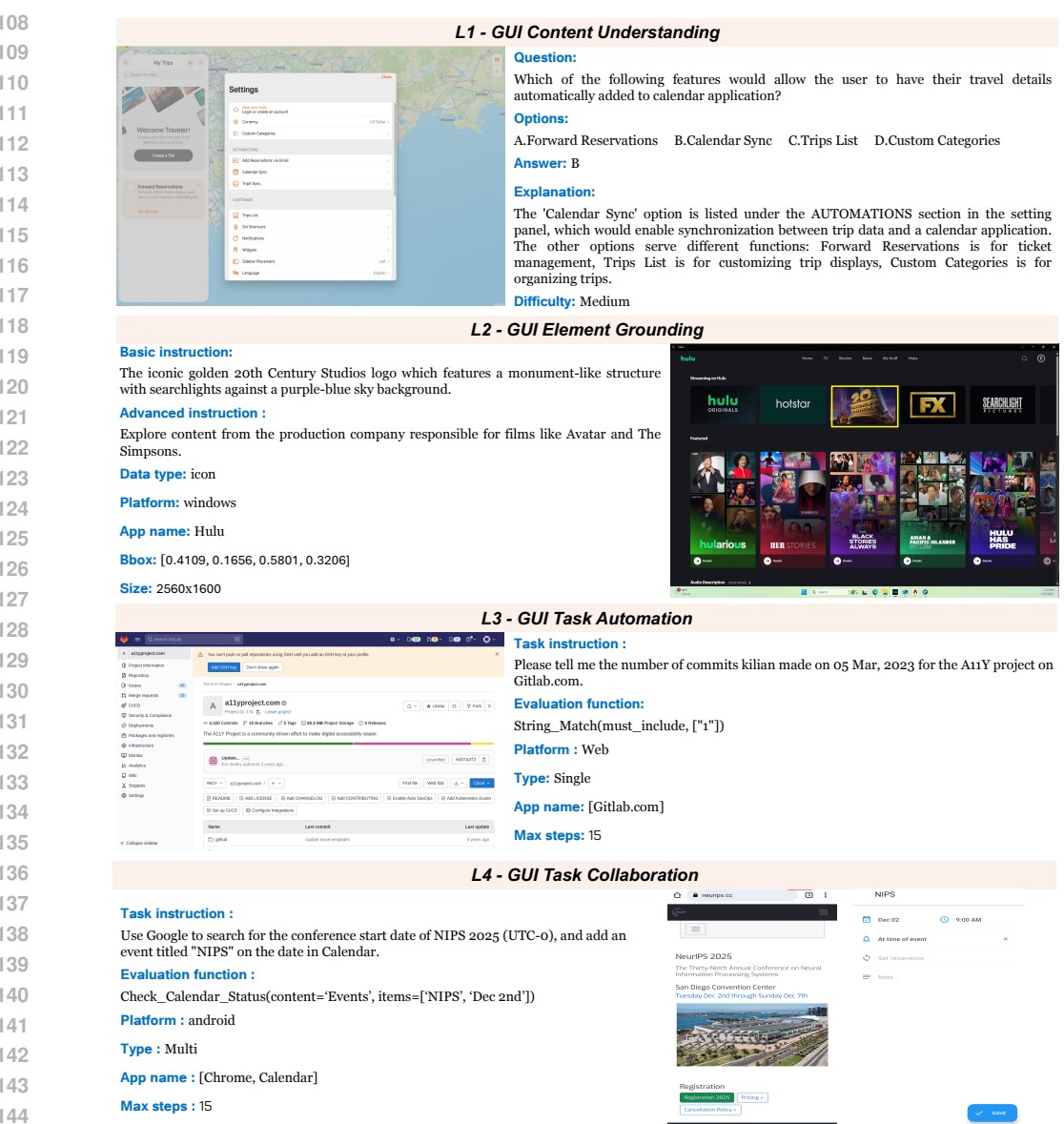

Figure 2: **Examples for all levels.** Both of L1 and L2 are offline tasks for quick evaluation. Tasks of L3 and L4 are evaluated in the virtual environment with an online manner.

## 2.2 L1-GUI CONTENT UNDERSTANDING

To reliably complete automated tasks, GUI agents must couple domain knowledge with visual observations, allowing them to interpret interface layouts, functionalities, and embedded information. This process is inherently challenging due to substantial variability in design paradigms across platforms, inconsistencies in interface conventions among applications within the same platform, and fragmented domain knowledge for specialized software. These complexities highlight the necessity of advanced perception and understanding mechanisms. However, the absence of comprehensive benchmarks has left such capabilities insufficiently and implicitly evaluated.

To address this gap, we introduce the first-level task: L1-GUI Content Understanding. Positioned at the foundation, this task emphasizes that GUI comprehension is a prerequisite for accomplishing any subsequent objectives.

**Task Definition.** At this task level, the objective is to evaluate an agent's ability to extract, interpret, and reason over information contained in GUI screenshots, without focusing on fine-grained element

localization or concrete operational actions. To this end, we formulate the evaluation as a Multiple-Choice Question Answering (MCQA) task grounded in visual observations, providing a standardized and quantifiable framework that simplifies assessment. Formally, the task can be defined as

$$o^* = \text{Agent}(\mathbf{V}, q, \mathcal{O}) \tag{1}$$

where $\mathbf{V}$ denotes the visual observation (GUI screenshot) presented to the agent, $q$ represents the question about the observation to evaluate comprehension, and $\mathcal{O} = \{o_1, o_2, \ldots, o_k\}$ represents the set of $k$ candidate options for question $q$, among which only one $o$ can correctly answer $q$.

The agent's goal is to analyze the question $q$, identify relevant information within $\mathbf{V}$, conduct reasoning, and finally select an option $o^*$ from $\mathcal{O}$ as the predicted answer. The key to achieving this goal is the proper construction of the pair $(q, \mathcal{O})$, as the effectiveness of the evaluation depends on both the quality of the question and the relevance of the options. Therefore, we argue that a diverse and sufficiently large set of well-constructed $(q, \mathcal{O})$ pairs about GUI elements and operations, with varying levels of difficulty, is essential to effectively evaluate the GUI understanding capabilities.

**Data Collection and Annotation.** We constructed a diverse dataset of application and website screenshots and derived high-quality Question–Options–Answer pairs through a multi-stage pipeline combining LLM generation, verification, and manual inspection. Examples are shown in Figure 2, and the detailed data acquisition process and annotation protocol are provided in Appendix A.4.1.

**Evaluation Metrics.** For each question, we adopt accuracy as the evaluation metric, consistent with common QA tasks. In addition, we propose a refined weighted accuracy that more rigorously considers the number of options; full details are presented in Appendix A.5.1.

## 2.3 LEVEL 2: GUI ELEMENT GROUNDING

Accurate grounding of interactive UI elements is a fundamental prerequisite for executing GUI-based tasks, requiring agents to localize target elements conditioned on task objectives and visual observations. Despite recent progress, several challenges persist: (1) visual ambiguity from highly similar elements, such as identical icons or buttons; (2) dynamic disruptions from pop-ups or transient notifications; and (3) the difficulty of distinguishing inactive from active regions. As grounding directly determines reliability, addressing these issues is critical for effective GUI automation.

However, existing benchmarks remain limited. ScreenSpot (Cheng et al., 2024; Wu et al., 2024b) is nearly saturated, while ScreenSpot Pro (Li et al., 2025) is restricted to narrow domains. More importantly, benchmark instructions are often overly direct (e.g., "submit the paper"), failing to capture the nuanced references and reasoning required in real-world tasks. This mismatch leaves a substantial gap between benchmark evaluation and practical deployment. To bridge this gap, we introduce a new dataset that expands domain coverage and incorporates more diverse, realistic instructions. By systematically categorizing instructions by descriptive type, our benchmark better reveals model weaknesses and aligns evaluation with real-world agent reasoning.

**Task Definition.** Accurate perception and understanding by an agent typically require validation through concrete actions, analogous to human interactions with GUI elements, to execute subsequent task steps. Building upon the comprehension capabilities assessed in L1, we propose *L2-GUI Element Grounding* to further measure the agent's spatial localization ability, specifically, the accurate identification of actionable GUI elements. This ability aligns precisely with the requirements of a grounding task, formally defined as:

$$p = \text{Agent}(\texttt{ins}, \mathbf{V}) \tag{2}$$

where $\texttt{ins}$ represents an instruction for the GUI element to be localized, which can be derived from a direct user task or the agent's internal reasoning process. The output $p$ denotes the resulting location of the target element, typically represented by the coordinates $(x, y)$ that indicate the activation point of the interactive element. The definition of $\texttt{ins}$ constitutes the core component of this level. In the context of GUI tasks, the description of an element can encompass various attributes, including appearance, approximate spatial position, and functionality.

**Data Collection and Annotation.** We extend the data from L1 through annotating interactive elements and generating grounding instructions with a multistep procedure, enabling multidimensional

analysis on a consistent data foundation. An example of L2 and the details of this design are provided in Figure 2 and Appendix A.4.2, respectively.

**Evaluation Metrics.** Following the evaluation protocol of ScreenSpot (Cheng et al., 2024), we computed accuracy separately for the Basic and Advanced instruction types. For each interactive element, a prediction was considered successful if the agent's predicted point of interaction–represented as a coordinate $(x, y)$–fell within the annotated bounding box. Otherwise, it was marked as a failure. The final accuracy was calculated as the proportion of successful predictions over the total number of evaluated elements.

## 2.4 LEVEL 3: GUI TASK AUTOMATION

To accomplish tasks within a single application, agents must combine interface comprehension and precise element localization with advanced planning and dynamic reasoning. A typical workflow involves parsing task instructions, perceiving interface content, grounding target components, and decomposing high-level objectives into executable actions (e.g., clicks, typing). The agent then iteratively interacts with the environment, adjusting its strategy based on real-time feedback. This tightly integrated cycle of perception, decision-making, and interaction underpins robust task completion, particularly in multi-step scenarios.

Key challenges include resolving ambiguous instructions, handling dynamic UI changes such as pop-ups or context shifts, and efficiently planning long action sequences. Despite their prevalence in real-world automation, these abilities are rarely assessed systematically across platforms. To fill this gap, we propose L3-GUI Task Automation as the third level of our benchmark, targeting end-to-end automation within a single, potentially complex, application. This level serves as a critical bridge between low-level perception/grounding and higher-level, generalizable task-solving skills.

**Task Definition.** We formally define this level as follows: The agent is required to complete a multi-step task within a single application by generating a sequence of actions that directly manipulate the user interface to fulfill a specified objective. At each time step $t$, the agent receives a visual observation $\mathbf{V}_t$ of the current UI state and generates an action $A_t$ with corresponding parameters $P_t$, based on the task instruction ins, the history $\mathcal{H}_t$, and involved applications $\mathcal{S}$ (with $\mathcal{S} \in \{\text{App}_1, \text{App}_2, \ldots, \text{App}_n\}$ for single-app scenarios). The process is formally described as:

$$
\begin{aligned}
A_t, P_t &= \text{Agent}(\text{ins}, \mathbf{V}_t, \mathcal{H}_t, \mathcal{S}_s) \\
\mathbf{V}_{t+1} &= \text{Env}(A_t, P_t) \\
\mathcal{H}_{t+1} &= \{\mathcal{H}_t, (V_t, A_t, P_t)\}
\end{aligned}
\tag{3}
$$

Here, $\mathcal{H}_t$ denotes the contextual history, which normally consists of previous observations and action sequences. In practice, the implementation of history typically follows two styles. The first style encapsulates the entire interaction process within multi-turn dialogues, while the second one condenses history into natural language and injects it into the prompt. The agent-environment interaction proceeds iteratively until a maximum number of steps ($t = T_{\max}$) is reached or a terminal action ($A_t \in [\text{FINISH, FAIL}]$) is predicted.

**Task Collection and Curation.** Our benchmark integrates tasks from multiple established platforms and further introduces a curated macOS task suite (MacOSArena) to ensure comprehensive and realistic coverage. Detailed task sources, review protocols, and the design of MacOSArena are provided in Appendix A.4.3. We also provide an example in Figure 2.

**Evaluation Metrics.** An ideal GUI agent should demonstrate both accuracy and efficiency. Yet, existing benchmarks rely almost exclusively on Success Rate (SR), overlooking how efficiently tasks are accomplished. To fill this gap, we introduce the Efficiency–Quality-Aware (EQA) metric, which jointly accounts for task success and action economy, rewarding agents that complete more tasks with fewer steps. **We elaborate the defination and formulation of EQA in Appendix A.5.2**.

## 2.5 LEVEL 4: GUI TASK COLLABORATION

Real-world automation often requires agents to coordinate actions across multiple applications, managing heterogeneous interfaces and interdependent subtasks. Such scenarios demand more than

localized planning: agents must adopt a global perspective, track cross-application dependencies, sequence operations coherently, and maintain information flow. They must also demonstrate long-horizon reasoning, robust error recovery, adaptability to interface changes, and resilience to runtime variability-capabilities that remain challenging for contemporary GUI agents.

Despite the importance of collaboration and global reasoning, existing benchmarks rarely evaluate these aspects systematically. To bridge this gap, we introduce L4-GUI Task Collaboration as the fourth level of our benchmark, designed to assess agents' ability to perform reasoning, coordination, and adaptive automation across multiple applications.

**Task Definition.** Extending the formulation above, L4 evaluates the agent's ability to coordinate complex workflows involving multiple applications. The agent must generate and execute a sequence of actions that may interact with any application in the set $\mathcal{S}_m$, where $\mathcal{S}_m$ represents a subset of $k$ applications selected from the available pool, i.e., $\mathcal{S}_m \subseteq \{\mathtt{App}_1, \mathtt{App}_2, \ldots, \mathtt{App}_N\}$ with $|\mathcal{S}_m| = k$, to accomplish a collaborative high-level task. Formally, the agent-environment loop in Equation 3 changes as follows:

$$A_t, P_t = \mathrm{Agent}(\mathtt{ins}, \mathbf{V}_t, \mathcal{H}_t, \mathcal{S}_m)$$
$$\mathbf{V}_{t+1} = \mathrm{Env}(A_t, P_t) \tag{4}$$
$$\mathcal{H}_{t+1} = \{\mathcal{H}_t, (V_t, A_t, P_t)\}$$

Meanwhile, $\mathcal{H}_t$ now aggregates the interaction history across all relevant app environments. The process terminates when either the step limit is reached or a terminal action is predicted.

**Task Collection and Design.** We further extend L3 tasks to L4 workflows as shown in Figure 2, capturing inter-app coordination and realistic cross-interface scenarios. Detailed task construction principles are provided in Appendix A.4.4.

**Evaluation Metrics.** We adopt the same evaluation metrics as in L3, i.e., SR and EQA. For both levels, the completion result is determined by verifying the final state and counting the number of steps taken, without the need to consider the individual states of multiple applications in $\mathcal{S}_m$.

## 3 BENCHMARKING GUI AGENT BASELINES

In this section, we evaluate a diverse set of contemporary VLMs and LLMs, covering both open-source and proprietary models, on MMBench-GUI to provide a comprehensive view of current GUI-agent performance. Each method is supplied only with screenshots and task descriptions, deliberately excluding auxiliary artifacts such as accessibility (A11y) trees or Set-of-Marks (SoM) data, thereby aligning more closely with real-world deployment. We evaluate a broad range of both proprietary and open-source models. The proprietary models include GPT-4o, Claude-3.7, Qwen-Max-VL (Hurst et al., 2024; Anthropic, 2025; Bai et al., 2023) and the open-sourced models include Qwen2.5 series, UI-TARS series, InternVL series, Aguvis, ShowUI, UGround, OS-Atlas (Bai et al., 2025; Qin et al., 2025; Zhu et al., 2025; Xu et al., 2024b; Lin et al., 2024; Gou et al., 2024; Wu et al., 2024b). We also assess hybrid configurations such as GPT-4o+UGround. The benchmarking details are provided in the Appendix A.7.

### 3.1 BENCHMARK RESULTS ON L1-GUI CONTENT UNDERSTANDING

Table 1 reports model performance on the GUI Understanding task (L1) across three difficulty levels and six platforms. InternVL3-72B consistently achieves the highest scores on all conditions, with Qwen2.5-VL-72B and Qwen-Max-VL following. GPT-4o shows moderate results, while the Claude variants and UI-TARS-72B-DPO perform substantially worse.

Three consistent patterns emerge. First, performance declines as task difficulty increases, with Easy always outperforming Medium and Hard. Second, cross-platform variability is evident: macOS and Linux generally yield higher scores, whereas Android and Web introduce larger fluctuations and lower accuracy. Third, InternVL3-72B not only maintains the top rank (79.2%, 77.9%, 75.7% on Easy, Medium, and Hard, respectively) but also exhibits the smallest performance drop, indicating stronger robustness. Qwen2.5-VL-72B remains second, while GPT-4o degrades more sharply on harder items. In summary, these results reveal substantial differences in model capability for GUI content understanding, offering a solid quantitative foundation for the deeper analysis presented in the following section.

Table 1: **Performance on L1-GUI Content Understanding.** 'Overall' represents the aggregated score across all platforms, calculated as a weighted sum of individual platform scores.

| Model | Windows | MacOS | Linux | iOS | Android | Web | Overall |
|---|---|---|---|---|---|---|---|
| | | | *Easy Level* | | | | |
| GPT-4o (2024) | 62.47 | 67.89 | 62.38 | 58.52 | 56.41 | 58.51 | 60.16 |
| Claude-3.5 (2024) | 41.34 | 50.04 | 41.61 | 42.03 | 38.96 | 41.79 | 41.54 |
| Claude-3.7 (2025) | 34.66 | 49.05 | 39.37 | 42.76 | 37.45 | 40.80 | 39.08 |
| Qwen-Max-VL (2023) | 69.05 | 72.51 | 69.91 | 70.82 | 63.09 | 69.46 | 68.15 |
| Qwen2.5-VL-72B (2025) | 65.86 | 75.23 | 73.02 | 67.24 | 58.09 | 72.08 | 66.98 |
| UI-TARS-72B-DPO (2025) | 41.59 | 28.52 | 35.16 | 31.08 | 52.25 | 35.33 | 40.18 |
| InternVL3-72B (2025) | **74.67** | **78.72** | **79.16** | **83.57** | **80.10** | **81.18** | **79.15** |
| | | | *Medium Level* | | | | |
| GPT-4o (2024) | 56.33 | 63.13 | 59.70 | 54.06 | 57.69 | 54.98 | 57.24 |
| Claude-3.5 (2024) | 39.28 | 47.63 | 45.97 | 44.57 | 42.03 | 34.33 | 41.26 |
| Claude-3.7 (2025) | 39.34 | 39.23 | 42.28 | 39.45 | 36.05 | 36.17 | 38.39 |
| Qwen-Max-VL (2023) | 63.40 | 73.85 | 66.90 | 68.02 | 63.66 | 64.59 | 65.44 |
| Qwen2.5-VL-72B (2025) | 66.29 | 72.73 | 72.63 | 59.27 | 66.24 | 68.24 | 67.45 |
| UI-TARS-72B-DPO (2025) | 38.83 | 41.60 | 37.14 | 41.72 | 54.74 | 31.55 | 41.77 |
| InternVL3-72B (2025) | **71.46** | **78.58** | **79.88** | **78.43** | **81.36** | **78.67** | **77.89** |
| | | | *Hard Level* | | | | |
| GPT-4o (2024) | 60.69 | 60.38 | 52.42 | 45.27 | 50.93 | 50.83 | 53.49 |
| Claude-3.5 (2024) | 37.40 | 42.70 | 34.07 | 40.86 | 36.96 | 38.11 | 37.55 |
| Claude-3.7 (2025) | 32.99 | 34.48 | 31.97 | 39.20 | 36.99 | 38.92 | 35.65 |
| Qwen-Max-VL (2023) | 66.64 | 67.59 | 65.80 | 60.23 | 58.78 | 65.34 | 63.69 |
| Qwen2.5-VL-72B (2025) | 70.68 | 68.91 | 70.98 | 57.59 | 53.94 | 68.10 | 64.56 |
| UI-TARS-72B-DPO (2025) | 31.48 | 35.87 | 24.19 | 36.33 | 58.13 | 19.94 | 35.78 |
| InternVL3-72B (2025) | **75.08** | **77.44** | **76.19** | **70.37** | **75.73** | **78.11** | **75.70** |

Table 2: **Performance on the L2-GUI Element Grounding.** "Adv." stands for advanced, while "Avg." refers to the weighted average of all results in a row, where the weights correspond to the proportion of tasks for each platform and mode relative to the total number of tasks.

| Model | Windows | | MacOS | | Linux | | iOS | | Android | | Web | | Avg |
|---|---|---|---|---|---|---|---|---|---|---|---|---|---|
| | Basic | Adv. | Basic | Adv. | Basic | Adv. | Basic | Adv. | Basic | Adv. | Basic | Adv. | |
| GPT-4o (2024) | 1.5 | 1.1 | 8.7 | 4.3 | 1.1 | 1.0 | 5.1 | 3.3 | 2.5 | 1.4 | 3.2 | 2.9 | 2.9 |
| Claude-3.7 (2025) | 1.5 | 0.7 | 12.5 | 7.5 | 1.1 | 0.0 | 13.7 | 10.6 | 1.4 | 1.4 | 3.2 | 2.3 | 4.7 |
| Qwen-Max-VL (2023) | 43.9 | 36.7 | 58.8 | 56.1 | 53.9 | 30.1 | 77.4 | 59.1 | 79.5 | 70.1 | 74.8 | 58.8 | 58.0 |
| Aguvis-7B-720P (2024b) | 37.3 | 21.7 | 48.1 | 33.3 | 33.5 | 25.0 | 67.5 | 65.2 | 61.00 | 51.0 | 61.6 | 45.5 | 45.7 |
| ShowUI-2B (2024b) | 9.2 | 4.4 | 24.1 | 10.4 | 25.1 | 11.7 | 29.0 | 19.7 | 17.4 | 8.7 | 22.9 | 12.7 | 16.0 |
| OS-Atlas-Base-7B (2024b) | 36.9 | 18.8 | 44.4 | 21.7 | 31.4 | 13.3 | 74.8 | 48.8 | 69.6 | 46.8 | 61.3 | 35.4 | 41.4 |
| UGround-V1-7B (2024) | 66.8 | 39.0 | 71.3 | 48.6 | 56.5 | 31.1 | 92.7 | 70.9 | 93.5 | 71.0 | 88.7 | 64.6 | 65.7 |
| InternVL3-72B (2025) | 70.1 | 42.6 | 75.7 | 52.3 | 59.2 | 41.3 | 93.6 | 80.6 | 92.7 | 78.6 | 90.7 | 65.9 | 72.2 |
| Qwen2.5-VL-72B (2025) | 55.7 | 33.8 | 49.9 | 30.1 | 40.3 | 20.9 | 56.1 | 28.2 | 55.6 | 25.4 | 68.4 | 45.8 | 41.8 |
| Qwen2.5-VL-7B (2025) | 31.4 | 16.5 | 31.3 | 22.0 | 21.5 | 12.2 | 66.6 | 55.2 | 35.1 | 35.2 | 40.3 | 32.5 | 33.9 |
| UI-TARS-1.5-7B (2025) | 68.3 | 39.0 | 69.0 | 44.5 | 64.4 | 37.8 | 88.5 | 69.4 | 90.5 | 69.3 | 81.0 | 56.5 | 64.3 |
| UI-TARS-72B-DPO (2025) | **78.6** | **51.8** | **80.3** | **62.7** | **68.6** | **51.5** | 90.7 | **81.2** | 93.0 | 80.0 | 88.1 | **68.5** | **74.3** |

## 3.2 BENCHMARK RESULTS ON L2-GUI ELEMENT GROUNDING

Table 2 presents the results for the L2 element grounding task across six platforms and two instruction types. Performance varies widely: GPT-4o and Claude-3.7 exhibit almost no grounding ability, whereas open-source models such as UI-TARS-72B-DPO, InternVL3-72B, UGround-V1-7B, and Qwen2.5-VL-72B achieve substantially higher scores. The strongest results are obtained by UI-TARS-72B-DPO (74.25%) and InternVL3-72B (72.20%), both showing robust cross-platform consistency, for instance, UI-TARS surpasses 80% on macOS, Android, and Web in the Basic setting while remaining competitive on Linux and iOS. A clear platform-dependent pattern emerges: high-performing models achieve greater accuracy on mobile and web environments (e.g., UI-TARS 93.54% on Android and 88.71% on Web) than on desktop systems. Furthermore, all models show

Table 3: **Evaluation result of L3-GUI Task Automation and L4-GUI Task Collaboration.** Values in **bold** indicate the highest score within each group; underlined values indicate the second highest.

| Model | Windows | | MacOS | | Linux | | Android | | Web | | Avg | |
|---|---|---|---|---|---|---|---|---|---|---|---|---|
| | SR | EQA | SR | EQA | SR | EQA | SR | EQA | SR | EQA | SR | EQA |
| L3-GUI Task Automation (Max Steps=15) | | | | | | | | | | | | |
| GPT-4o (2024) | 5.6 | 3.3 | 0.0 | 0.0 | 6.8 | 4.4 | 19.0 | 8.9 | 1.9 | 1.5 | 7.2 | 4.1 |
| Claude-3.7 (2025) | 6.8 | 3.4 | 8.6 | 2.8 | 7.4 | 4.2 | 11.2 | 3.5 | 1.9 | 1.5 | 6.8 | 3.4 |
| Aguvis-72B (2024b) | 4.1 | 2.0 | 0.0 | 0.0 | 3.1 | 1.6 | 18.1 | 10.8 | 5.8 | 2.2 | 6.2 | 3.2 |
| UI-TARS-7B (2025) | 6.2 | 3.2 | 2.9 | 2.2 | 16.5 | 13.4 | 23.7 | 15.5 | 11.0 | 6.1 | 13.7 | 9.6 |
| UI-TARS-1.5-7B (2025) | 11.1 | 6.0 | **11.4** | 6.6 | 26.5 | 18.7 | 30.2 | 17.9 | 20.8 | 12.2 | 22.0 | 14.0 |
| UI-TARS-72B-DPO (2025) | 11.1 | 5.4 | **11.4** | 7.8 | 30.3 | 18.9 | 43.1 | 26.6 | 23.4 | 17.1 | **26.1** | **16.5** |
| Qwen2.5-VL-72B (2025) | 11.8 | 7.2 | 2.9 | 2.0 | 9.8 | 5.4 | 16.4 | 9.8 | 15.6 | 9.9 | 12.2 | 7.3 |
| GPT-4o + UGround-V1-7B (2024) | 13.1 | 8.1 | 2.9 | 1.0 | 16.1 | 8.7 | 34.5 | 21.1 | 23.2 | 16.7 | 19.4 | 11.9 |
| GPT-4o + UI-TARS-1.5-7B (2025) | **14.5** | 6.8 | 2.9 | 0.9 | 20.2 | 11.1 | 33.6 | 15.2 | **28.6** | **17.5** | 22.2 | 11.8 |
| L3-GUI Task Automation (Max Steps=50) | | | | | | | | | | | | |
| GPT-4o (2024) | 3.5 | 2.3 | 2.9 | 1.7 | 11.6 | 9.1 | 21.6 | 10.8 | 3.3 | 3.2 | 9.4 | 6.3 |
| Claude-3.7 (2025) | 6.4 | 4.0 | **11.4** | 4.2 | 10.3 | 6.3 | 11.2 | 3.6 | 2.6 | 2.6 | 8.1 | 4.5 |
| Aguvis-72B (2024b) | 3.5 | 1.6 | 0.0 | 0.0 | 4.2 | 2.0 | 19.8 | 14.7 | 9.0 | 3.8 | 7.4 | 4.3 |
| UI-TARS-7B (2025) | 8.4 | 6.1 | 2.9 | 2.5 | 23.9 | 13.9 | 25.4 | 18.1 | 13.6 | 8.0 | 17.8 | 11.2 |
| UI-TARS-1.5-7B (2025) | 15.9 | 11.3 | **11.4** | 7.0 | 29.8 | 21.3 | 31.6 | 22.2 | 26.0 | 17.5 | 25.6 | 17.9 |
| UI-TARS-72B-DPO (2025) | 17.9 | 11.8 | **11.4** | 8.4 | 31.4 | 25.4 | 45.7 | 35.2 | 21.4 | 16.9 | **27.9** | **21.6** |
| Qwen2.5-VL-72B (2025) | 9.7 | 6.9 | 5.7 | 4.0 | 10.6 | 7.9 | 27.6 | 21.8 | 14.4 | 9.7 | 13.7 | 10.1 |
| GPT-4o + UGround-V1-7B (2024) | 20.7 | 11.9 | 5.7 | 3.2 | 19.5 | 10.9 | 47.4 | 37.2 | 26.5 | 25.5 | 25.1 | 17.5 |
| GPT-4o + UI-TARS-1.5-7B (2024) | **26.2** | **17.3** | 8.6 | 5.0 | 22.9 | 13.8 | 42.2 | 33.1 | **28.6** | 21.0 | 27.2 | 18.7 |
| L4-GUI Task Collaboration (Max Steps=15) | | | | | | | | | | | | |
| GPT-4o (2024) | 7.5 | 5.9 | 0.0 | 0.0 | 3.5 | 2.4 | 0.0 | 0.0 | 2.1 | 0.1 | 3.0 | 2.2 |
| Claude-3.7 (2025) | 3.6 | 1.6 | 2.9 | **2.0** | 7.3 | 4.8 | 0.0 | 0.0 | – | – | 4.8 | 3.0 |
| Aguvis-72B (2024b) | 3.2 | 3.0 | 0.0 | 0.0 | 1.6 | 0.4 | 3.3 | 3.2 | – | – | 1.9 | 1.2 |
| UI-TARS-7B (2025) | 3.2 | 3.0 | 0.0 | 0.0 | 4.0 | 2.2 | 6.7 | 9.6 | – | – | 3.6 | 3.1 |
| UI-TARS-1.5-7B (2025) | 3.2 | 3.0 | 2.9 | 0.8 | 5.0 | 4.0 | 6.7 | 6.6 | – | – | 4.6 | 3.7 |
| UI-TARS-72B-DPO (2025) | 3.2 | 3.1 | **5.7** | 1.5 | 7.5 | 5.9 | 10.0 | 9.6 | – | – | 6.8 | 5.2 |
| Qwen2.5-VL-72B (2025) | 6.2 | 4.2 | 0.0 | 0.0 | 2.5 | 1.7 | 6.7 | 6.1 | – | – | 3.4 | 2.5 |
| GPT-4o + UGround-V1-7B (2024) | 9.3 | 5.1 | 0.0 | 0.0 | 3.6 | 2.5 | 3.3 | 3.2 | – | – | 3.9 | 2.6 |
| GPT-4o + UI-TARS-1.5-7B (2025) | **12.3** | **6.4** | 0.0 | 0.0 | 5.6 | 3.8 | 23.3 | 21.1 | 4.3 | 0.7 | 8.4 | 6.2 |
| L4-GUI Task Collaboration (Max Steps=50) | | | | | | | | | | | | |
| GPT-4o (2024) | 6.2 | 5.0 | 0.0 | 0.0 | 5.9 | 5.4 | 0.0 | 0.0 | 2.1 | 1.5 | 4.0 | 3.5 |
| Claude-3.7 (2025) | 6.3 | 4.8 | 2.9 | 2.0 | **9.3** | **7.4** | 0.0 | 0.0 | – | – | 5.5 | 3.9 |
| Aguvis-72B (2024b) | 3.2 | 3.1 | 0.0 | 0.0 | 1.6 | 0.4 | 6.7 | 6.4 | – | – | 2.4 | 1.7 |
| UI-TARS-7B (2025) | 6.2 | 5.2 | 0.0 | 0.0 | 4.6 | 3.6 | 10.0 | 6.5 | – | – | 4.9 | 3.7 |
| UI-TARS-1.5-7B (2025) | 6.2 | 6.0 | 2.9 | 0.9 | 7.6 | 5.5 | 13.3 | 13.1 | – | – | 7.4 | 5.9 |
| UI-TARS-72B-DPO (2025) | 9.3 | 6.2 | **5.7** | 2.3 | 8.5 | 7.2 | 20.0 | 11.8 | – | – | **9.8** | 6.9 |
| Qwen2.5-VL-72B (2025) | 6.2 | 5.2 | 0.0 | 0.0 | 1.6 | 1.3 | 6.7 | 6.5 | – | – | 2.9 | 2.8 |
| GPT-4o + UGround-V1-7B (2024) | 9.3 | 5.0 | 0.0 | 0.0 | 5.5 | 3.8 | 6.7 | 6.4 | – | – | 5.3 | 3.7 |
| GPT-4o + UI-TARS-1.5-7B (2024) | **12.3** | 6.8 | 2.9 | 1.0 | 7.5 | 5.6 | 23.3 | 21.7 | 4.1 | 2.0 | 9.8 | 7.4 |

reduced accuracy under Advanced instructions compared to Basic ones, highlighting the additional challenge posed by abstract or functional cues.

### 3.3 BENCHMARK RESULTS ON L3/L4-GUI TASK AUTOMATION/COLLABORATION

Tables 3 present results for single-app (L3) and multi-app (L4) automation under step limits of 15 and 50. A unified evaluation pipeline was applied, with standardized prompts and action spaces for general-purpose models and official settings for GUI-specific agents.

On L3 tasks, overall performance remains modest: the best method, GPT-4o + UI-TARS-1.5-7B, achieves only 26.6% SR, while most models fall below 20%. GUI-specific agents such as UI-TARS-

72B-DPO outperform others, particularly on Linux and Android, whereas language-centric models like GPT-4o and Claude-3.7 perform poorly. Crucially, combining general-purpose models with grounding modules consistently improves results—for example, GPT-4o alone yields 6.13% SR, but surpasses 17% when paired with UGround or UI-TARS. In contrast, L4 performance drops sharply: the top system reaches just 8.78% SR, with most models under 6%, underscoring the difficulty of cross-application automation. Allowing longer action sequences (50 vs. 15 steps) improves SR and EQA across settings but does not close the gap, suggesting persistent challenges in long-horizon planning and multi-step execution.

Platform differences are also pronounced. Android and Web yield relatively higher scores (e.g., GPT-4o + UI-TARS-1.5-7B achieves 33.10%/25.81% SR/EQA on Android), whereas desktop environments, especially macOS, lag substantially.

Overall, these results highlight the benefits of integrating planning with grounding for L3 tasks, while revealing a substantial performance gap in L4 scenarios that exposes the limitations of current agents in robust, long-horizon, multi-application automation.

# 4 ANALYSIS OVERVIEW

Synthesizing the results in Section 3 across L1–L4 yields six findings that delineate the current limits of GUI agents:

- **Planning is not enough**: general-purpose LMs plan well but miss fine-grained interactions; coupling planners with specialized grounders reliably lifts performance.

- **Accurate visual grounding is the primary driver of success**: improvements in localization translate directly into higher SR and more stable behavior.

- **Efficiency remains underexplored**: our EQA metric reveals pervasive step redundancy; principled early stopping and step-aware policies matter.

- **Action-space bottlenecks**: many failures arise from missing or overly coarse primitives, not perception alone—richer, better-scoped actions are required.

- **Fragility under complexity and dynamics**: accuracy and efficiency drop with instruction abstraction, UI volatility, and longer horizons, indicating limited generalization.

- **Cross-application gaps**: multi-app workflows fail mainly due to deficits in memory/state tracking and cross-app information flow rather than deficiencies in recognition per se.

Together, these findings chart a concrete path forward: pair strong planners with high-precision grounders, expand and normalize the action space, elevate efficiency to a first-class objective via EQA-aware policies, and equip agents with persistent memory for cross-application orchestration.

We provide complete analyses and evidence in Appendix A.6 with extended textual discussions and complementary figures and tables. These materials substantiate each finding and offer diagnostic guidance for future work.

# 5 CONCLUSION

In this work, we presented **MMBench-GUI**, a novel hierarchical multi-platform evaluation framework that comprehensively assesses the capabilities and limitations of GUI automation agents. Through rigorous evaluations across multiple operating systems and diverse tasks, we uncovered critical insights into key performance bottlenecks, particularly highlighting the importance of accurate visual grounding, sophisticated planning, and robust cross-platform generalization. Our findings demonstrate that modular architectures integrating specialized grounding modules significantly improve performance, addressing inherent limitations of general-purpose language models. Additionally, our analysis underscores the importance of improving long-horizon reasoning, adaptive error recovery, and effective memory and state management to address complex and ambiguous GUI scenarios. MMBench-GUI thus provides a foundational benchmarking resource and actionable guidance for future research efforts, advancing the development of robust, reliable, and practically applicable GUI automation agents.

ETHICS STATEMENT

Our work complies with the ICLR Code of Ethics. The proposed dataset is constructed without collecting any personally identifiable information or sensitive data. All screenshots, metadata information are obtained from synthetic or publicly accessible software environments and do not involve real users' private data. Human experts are limited to interface-level information (e.g., UI element labels, bounding boxes, or action descriptions) without exposure to personal content. The released resources (dataset, and code) are intended solely for research purposes to advance open and reproducible study of cross-platform computer use agents. We explicitly discourage any misuse of these resources in ways that could compromise privacy, security, or fairness. No conflicts of interest or sponsorship bias exist in this work, and all authors adhere to research integrity practices, including transparent documentation of data sources, collection procedures, and evaluation protocols

REPRODUCIBILITY STATEMENT

The reproducibility-critical aspects of our benchmark center on both the construction of the evaluation data and the model evaluation protocol. Accordingly, the appendix provides comprehensive descriptions of the data-construction methodology and the prompts used for labeling (Appendix A.4), together with a full account of the evaluation pipeline-covering inference setup (Appendix A.7), termination criteria, and metric computation (Appendix A.5), thereby enabling researchers to replicate the dataset and reproduce our evaluations with fidelity under transparent and repeatable conditions.

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

## A  APPENDIX

In this section, we present material that augments the main content, including more extensive analyses, formal derivations of the evaluation metrics, additional figures and tables, and a detailed exposition of the data-construction process. We strongly encourage readers to consult this section in parallel with the main paper, as doing so will facilitate a more comprehensive understanding and assessment of our contributions under space-constrained presentation in the body of the paper. The appendix is organized as follows:

- Appendix A.1: We explain the use of LLMs in our paper.
- Appendix A.2: We review related works in GUI Agent and corresponding benchmarks.
- Appendix A.3: We conduct a statistical analysis of the benchmark dataset.
- Appendix A.4: We elaborate the details about data collection, data annotation, task curation and task design.
- Appendix A.5: We formally define all evaluation metrics and detail their computation, including step-by-step formulas and measurement protocols.
- Appendix A.6: We present an in-depth analysis of the benchmark experiments and articulate several findings designed to inform subsequent work by the research community.
- Appendix A.7: We provide a detailed description of the evaluation setup and protocol.

### A.1  THE USE OF LARGE LANGUAGE MODELS (LLMS)

We affirm that this paper is prepared and written entirely by us. We did not use any large language models (LLMs) to generate the abstract, content, or any part of the text, The ideas, analysis, and conclusions presented are the sole product of the authors' original thought and research. We did, however, utilize LLMs to help us polish the writing of our paper.

In annotating the L1 and L2 datasets, we employed LLMs as annotators, consistent with other works. We subsequently conducted human screening and post-editing of the LLM outputs to ensure quality and to mitigate potential privacy concerns and inappropriate content.

### A.2  RELATED WORKS

**GUI Agents.** GUI agents have attracted growing interest, driven by advances like Anthropic's Computer-Use Agent[1] and OpenAI's Operator[2]. Currently, GUI Agents mainly fall into two paradigms: a) Modular agent schemes (Cheng et al., 2024; Gou et al., 2024; Yang et al., 2024; Zhang et al., 2025; Wu et al., 2025; Xie et al., 2025; Wang et al., 2025a), which typically employs general-purpose VLMs (*i.e.*, GPT-4o) as planners, integrated with a specially trained GUI grounding model for focused UI element localization; b) Native agent schemes (Xu et al., 2024b; Wu et al., 2024b; Lin et al., 2024; Sun et al., 2024; Qin et al., 2025; Yang et al., 2024), where planning and grounding are trained in an end-to-end manner. Modular approaches benefit from state-of-the-art components but face challenges in system-level alignment Cheng et al. (2024); Gou et al. (2024). In contrast, the native agent paradigm aligns capabilities more naturally during training (Wu et al., 2024b; Xu et al., 2024b; Qin et al., 2025). Both paradigms can use screenshots (Niu et al., 2024; Liu et al., 2024a), accessibility trees (A11y Trees) (Gao et al., 2023), and HTML pages (Furuta et al., 2023; Deng et al., 2023b) as input. However, A11y Trees and HTML codes vary across platforms, are prone to noise, and may cause excessive token length (Zheng et al., 2024; Hong et al., 2024; Cheng et al., 2024). Generally, in this work, we focus exclusively on the screenshot-only setting and propose a hierarchical, multi-platform benchmark to evaluate these vision-only native agents.

**GUI Benchmarks.** Effectively GUIs requires a sophisticated grasp of intertwined visual and textual cues, yet this complex domain remains largely outside the scope of general-purpose multimodal QA benchmarks (Liu et al., 2024c; Yue et al., 2024; Masry et al., 2022). While ScreenQA (Hsiao et al., 2022) and WebSRC (Chen et al., 2021) provide large-scale QA datasets based on Android screenshots and web pages respectively, and GUI-World introduces cross-platform GUI QA via video data, these

---

[1]https://www.anthropic.com/news/3-5-models-and-computer-use

[2]https://openai.com/index/computer-using-agent

Table 4: **Statistics of the evaluation data in MMBench-GUI.** Owing to the inherent restrictions of the iOS ecosystem, we were unable to include online tasks for iOS in L3&L4. All other platforms are covered in full.

| | Windows | MacOS | Linux | iOS | Android | Web | Overall |
|---|---|---|---|---|---|---|---|
| **L1** | L1 - Easy | | | | | | |
| | 271 | 84 | 196 | 115 | 307 | 221 | 1194 |
| | L1 - Medium | | | | | | |
| | 271 | 84 | 196 | 115 | 307 | 221 | 1194 |
| | L1 - Hard | | | | | | |
| | 271 | 84 | 196 | 115 | 307 | 221 | 1194 |
| **L2** | L2 - Basic | | | | | | |
| | 271 | 345 | 191 | 314 | 356 | 310 | 1787 |
| | L2 - Advanced | | | | | | |
| | 272 | 346 | 196 | 330 | 335 | 308 | 1787 |
| **L3** | 145 | 35 | 268 | - | 116 | 155 | 719 |
| **L4** | 35 | 35 | 101 | - | 30 | 47 | 248 |
| **Total** | **1536** | **1013** | **1344** | **989** | **1758** | **1483** | **8123** |

efforts offer limited support for interactive GUI agents. To evaluate visual grounding in GUI contexts, several benchmarks have emerged. ScreenSpot (Cheng et al., 2024) and its improved versions (Wu et al., 2024b; Li et al., 2025) support cross-platform UI grounding with progressively enhanced realism and annotation quality. UI-I2E-Bench (Liu et al., 2025a) and UI-Vision (Nayak et al., 2025) further expand this by aligning natural language instructions with GUI elements of varying scale and type. For reasoning and planning, offline benchmarks like (Rawles et al., 2023; Chen et al., 2024a; Li et al., 2024; Deng et al., 2023a; Kapoor et al., 2024; Lu et al., 2024) assess action prediction from fixed trajectories, while online benchmarks (Zhou et al., 2023; Xie et al., 2024; Bonatti et al., 2024; Rawles et al., 2024; Xu et al., 2024a; Liu et al., 2024b) enable interactive evaluation across platforms. However, macOS remains underexplored. Our MMBench-GUI benchmark addresses this gap by enabling online evaluation on macOS and emphasizing cross-platform robustness, providing a realistic and comprehensive evaluation for GUI agents.

### A.3 BENCHMARK STATISTICS

Table 4 enumerates the complete task inventory, 8123 distinct instances, broken down by operating platform, level, and difficulty band. Our benchmark has the following characteristics:

- L1-GUI Content Understanding (3 × QA splits). Each of the six platforms contributes an identical triplet of 271/84/196/115/307/221 items (Windows → Web), yielding 1194 examples per difficulty (Easy, Medium, Hard) and 3582 in total. This symmetry ensures that any performance gap across the three difficulty tiers cannot be attributed to data imbalance.

- L2-GUI Element Grounding (Basic vs. Advanced). The grounding set is roughly 50% larger than Level 1, with 1787 examples per split (Basic=Advanced). Note the deliberate platform skew: mobile platforms (iOS + Android = 686 or 38%) receive more queries than desktop platforms, reflecting the higher UI diversity and screen density of mobile apps.

- L3-GUI Task Automation (single application). A compact but varied set of 719 trajectories focuses on long-horizon planning within one application. Linux dominates (268 tasks) to capture the complexity of desktop productivity apps, while mobile splits are omitted for this level to avoid conflating OS diversity with task length.

- L4-GUI Task Collaboration (multiple applications). The hardest tier comprises 248 cross-application workflows. Although smaller, it intentionally spans all three desktop platforms and major mobile browsers (47 Web tasks, 30 Android tasks) to stress test memory hand-off and state persistence.

- **Aggregate balance.** Across the whole benchmark Windows (1536) and Android (1758) provide the two largest pools, but no single platform exceeds 22% of the corpus, guarding against model over-specialisation. The progressive shrinkage, from 3582 (L1) to 248 (L4), mirrors the increasing cost and difficulty of annotation, while still offering enough samples (about 250) for a statistically meaningful evaluation in the top tier.

Overall, the benchmark delivers (1) platform diversity, (2) controlled difficulty gradation, and (3) a realistic taper in task count that matches real-world annotation effort, thereby enabling fine-grained diagnosis of GUI agent capabilities at every competence level.

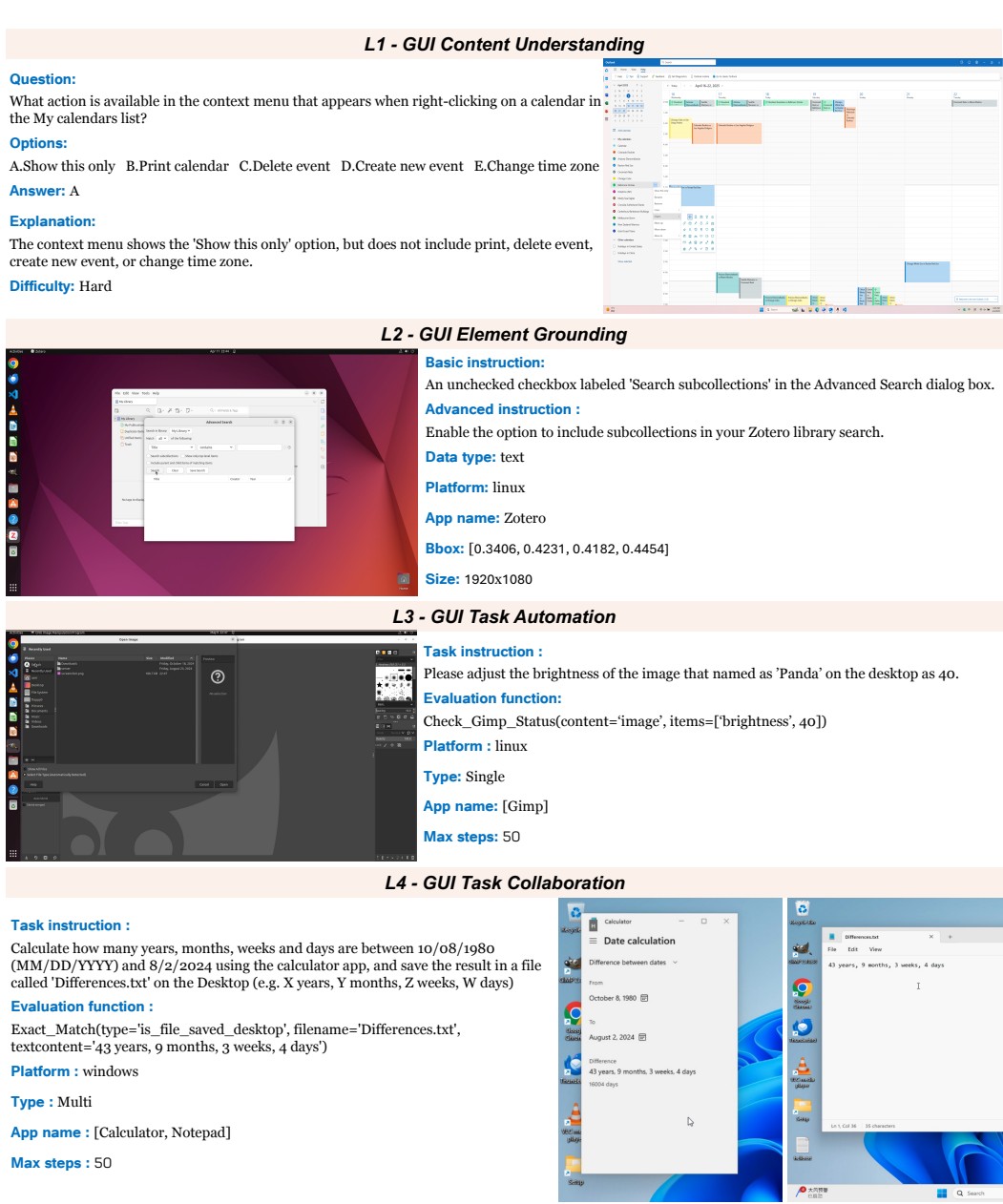

Figure 3: **More examples for all levels.**

### A.4 DATA&TASK COLLECTION AND ANNOTATION

#### A.4.1 L1-GUI CONTENT UNDERSTANDING

We manually collected screenshots from widely used applications and websites across all supported platforms, selected for their high usage frequency and representative user scenarios. In addition, we supplemented our data with a small number of screenshots sourced from publicly available datasets (Cheng et al., 2024; Li et al., 2025). To ensure diversity, we include screenshots of varying sizes, ranging from single-window to full-screen views, and accompanied each image with metadata; filenames were anonymized using an MD5-based encoding scheme constructed from a combination of platform, application name, and original file path, to avoid path conflicts and information leakage.

Then, we followed a four-step strategy to construct high-quality Question-Options-Answer pairs:

- Step 1: Claude 3.7 (Anthropic, 2025) was used to generate three questions for each image, corresponding to three levels of difficulty: easy, medium, and hard. Each question includes 4 to 6 answer options, with exactly one correct choice. In addition, Claude 3.7 was instructed to provide an explanation for each question, detailing the reasoning process that leads to the correct answer. In designing the questions for each image, we guided Claude 3.7 to focus on various aspects of the GUI, including the functionality of UI elements, structural relationships within the interface, content states, hierarchical layout, and executable tasks. The detailed prompt for this process is shown in Prompt 1.
- Step 2: We then used GPT-o4-mini (OpenAI, 2025) to verify the validity of each question, set of options, and answer, jointly considering the UI interface and the generated explanation. The errors were corrected with justification and revised explanations.
- Step 3: Then, GPT-o3 (OpenAI, 2025) was used to further review and refine the revised items following the same Prompt 2 as in Step 2.
- Step 4: Finally, manual sampling was performed to ensure overall quality and consistency.

By incorporating three different strong models across the pipeline, we reduced the risk of model-specific hallucinations and stylistic bias. We provide another example in Figure 3.

#### A.4.2 L2-GUI ELEMENT GROUNDING

We reuse the data from L1 to annotate additional agent capabilities, enabling multidimensional analysis on a consistent data foundation. This design facilitates exploration of inter-task correlations and addresses earlier research questions. We also manually labeled the positions of interactive elements, i.e. user-operable components such as buttons or icons, using bounding boxes, and categorized them as either Text or Icon, following the classification scheme used in ScreenSpot (Cheng et al., 2024).

We adopt a three-step procedure to generate grounding instructions for annotated interactive elements:

- Step 1: Claude 3.7 was prompted to produce two types of instruction per element: **Basic**, which describes visual features and approximate location to test perception-based grounding, and **Advanced**, which targets functional understanding through implicit cues. To increase diversity, three stylistic variants were generated for each type. The detailed prompt for this step is shown in Prompt 3.
- Step 2: We developed an annotation tool to manually review and refine these instructions, ensuring that each uniquely maps to a specific element.
- Step 3: A validated instruction per type was selected to form the final evaluation set.

#### A.4.3 L3-GUI TASK AUTOMATION

To ensure broad coverage and real-world relevance, our GUI task automation benchmark encompasses tasks across multiple major platforms, including Windows, Linux, macOS, web, and Android. Due to the inherent restrictions of the iOS ecosystem, iOS tasks are not currently included.

The majority of tasks are sourced from established public benchmarks, each of which leverages virtualization technology to provide robust and reproducible GUI environments. Specifically,

**Prompt 1:**

You are an expert GUI analyst for {os_name} and item-writer.

**Input:**

1. One screenshot of a GUI application.
2. The application's name ({app_name} or "Not available") — optional and for background only; do **not** mention it in any question text.

**Task:**

Create *exactly one* multiple-choice question about the screenshot at **each** of three difficulty levels (easy, medium, hard). For **every** question you generate:

- Write the stem in clear English that can be answered *only* by understanding the screenshot. Avoid trivial facts (e.g., "What color is the button?") unless color is functionally meaningful.
- Focus on tasks, labels, hierarchy, states, or affordances shown in the UI.
- Provide **4–6** answer options labeled "A", "B", "C", … in a JSON sub-object called `"options"`.
- Ensure **one and only one** option is strictly correct; the others must be clearly incorrect but plausible. Give the answer key (the letter of the correct option).
- Double-check yourself that the correct answer is indeed unique and unambiguous.
- Do **not** include the {app_name} or any other identifying text of the app in the stem or options.
- Give a concise `"explanation"` stating *why* the correct option is right and the others are not in 1–3 sentences.
- The hard question should require the answerer to think more about the screenshot, the question, and the options (you can also make options be easy to confuse).

**Output format:**

Return a single valid JSON array containing three objects (one per difficulty), in *English*, structured exactly like this schema:

```
[
  {
    "difficulty": "easy",
    "question": "<stem>",
    "options": {
      "A": "<option text>",
      "B": "<option text>",
      "C": "<option text>",
      "D": "<option text>"   // add "E","F" only if needed
    },
    "answer": "A",
    "explanation": "<brief rationale>"
  },
  ...
]
```

**Important Constraints:**

1. Produce only the JSON text—no markdown, headings, or commentary.
2. Validate that the JSON is syntactically correct before outputting.
3. After generation, internally review each Q&A for accuracy and compliance.

tasks for the Linux platform are drawn from OSWorld (Xie et al., 2024), Android from Android-World (Rawles et al., 2024), web from WebArenaLite-v2 (Liu et al., 2025b), and Windows from WindowsAgentArena (Bonatti et al., 2024). These resources have been extensively validated in prior research and collectively provide a diverse set of task scenarios. Importantly, our use of these benchmarks is not a simple replication. Each task underwent a rigorous manual review process, during which we excluded any instances likely to result in agent failure due to non-agent factors such as unstable network conditions, required account authentication, or platform-specific anomalies. This

**Prompt 2:**

You are a meticulous GUI-QA evaluator.

**Input:**

1. One screenshot (`image`) of a GUI application running on a {os_name}.
2. The application's name (`app_name`) – optional and strictly for background; *never* mention it in your output.
3. A JSON-like array (`qa_items`) containing three single-choice questions about the screenshot (intended levels: `easy`, `medium`, `hard`). Each object is expected to have the keys question, options, answer, difficulty, and optionally explanation.

\* Ignore cosmetic or syntactic issues in the supplied JSON (e.g., extra backticks, missing quotes, inconsistent key order, markdown fences).
\* Focus **only** on the content of `question`, `options`, and `answer` when deciding validity.

**Task:**

For each question, decide whether it is content-valid for use in a test. A question is valid only if all the following hold:

- The stem can be answered solely by inspecting the screenshot (no outside knowledge).
- Exactly one option is correct and that option is the one listed in `answer`.
- Incorrect options are clearly wrong yet still plausible.
- Neither stem nor options reveal the `app_name`.
- The difficulty label is reasonable (honor system; do not reject only for minor mislabelling).

The hard level should allow the answerer to think more deeply about the screenshot, the question, and the options. You may make the options easy to confuse.
\* Do **not penalise** minor formatting faults that do not affect the five substantive criteria.

**Output format:**

Return a JSON array of three objects in the original order, each with:

```
{
  "difficulty": "<same as input>",
  "valid": "yes" | "no",
  "comment": "<if valid: empty string; if not valid: brief reason
      why>",
  "fix": <if valid: null; if not valid: a *fully corrected* object
      that replaces the faulty one (same schema as above, with all
      issues fixed)>
}
```

**Notes:**

1. Provide an empty string (" ") for `comment` and "null" for `fix` when `valid` is "yes".
2. When `valid` is "no", supply both an actionable `comment` and a complete `fix` object that meets all criteria.
3. Do not wrap the result in markdown or add explanations outside the JSON.
4. Verify that the final JSON is syntactically correct before sending it.

curation ensures that the performance evaluations reflect true capabilities of the agent, rather than artifacts of the benchmarking environment.

To address the lack of existing online evaluation resources for the macOS platform, we introduce *MacOSArena*, a novel set of 70 curated tasks spanning 9 widely used macOS applications. Of these, 35 tasks are categorized as L3 tasks and the remaining 35 as L4 tasks. Task design for macOS follows the same principles as for other platforms, utilizing paired natural language instructions and screenshots to simulate virtual environments, thereby ensuring consistency and comparability across all platforms. This multi-platform, carefully curated task set provides a comprehensive and fair foundation for benchmarking GUI agents in realistic and heterogeneous settings. We provide extra illustrative examples in Figure 3 to demonstrate the details of L3 tasks.

**Prompt 3:**

You are a GUI agent currently operating on a {os_name}.

**Input:**

1. The first image is a screenshot from the {application} {app_or_web}, in which a selected element is highlighted with a distinctive red box and a red arrow.
2. The second image is the cropped region containing the selected element and corresponding box and arrow.
3. A simple and coarse description of the selected element.

**Task:**

Your task is to understand the possible role, function, and related global contextual information of the selected element on the current page from the first image. Then, from the second image, you can combine the global information from the first image to further analyze the relationship between the selected element and its surrounding information. The simple and coarse description can be regarded as a prior for the selected element. Finally, you are required to conclude two types of instructions for the selected element:

* *Basic Instruction*: Informative description that summarizes key information.
* *Advanced Instruction*: An indirect yet specific instruction that refers to the selected element.

**Guidelines for Generating Descriptions:**

*Basic Instruction*:

- Concise summary including appearance and position.
- Avoid referencing the red box or arrow.
- Examples:
  - "A circular icon with a white background and a magnifying glass symbol in black."
  - "Located in the top-right corner, to the right of the profile avatar icon."

*Advanced Instruction*:

- Focus on function and reasoning.
- Avoid visual/positional terms.
- Examples:
  - "Search some latest posts"
  - "Type in text to discover related content"

**Output format:**

Return a dictionary with:

```
{
    "basic_instruction": ["xxxx", "xxx", "xxx"],
    "advanced_instruction": ["xxxxx", "xxx", "xxx"]
}
```

**Notes:**

1. Ensure instructions are clear, unambiguous, and concise.
2. Do not mention the red box and arrow.
3. Coarse descriptions are only priors.

### A.4.4 L4-GUI TASK COLLABORATION

L4 tasks are designed as an extension of the single-app automation tasks in L3, with a primary focus on multi-application collaboration and information transfer across heterogeneous interfaces. For tasks in existing benchmarks that inherently involve multiple applications, we included them in our evaluation after a careful review of their availability and robustness. In addition, for those benchmarks lacking native multi-app workflows, we manually designed new tasks that explicitly require inter-app coordination. We also supplemented original multi-app tasks to further enrich the variety and complexity of cross-application scenarios.

A key design principle in constructing L4 tasks is to ensure that actions in one application provide necessary context or information for subsequent operations in another application. For example, in a

representative macOS task, the agent is required to search online for the time and location of CVPR 2023 and then create a corresponding event in the Calendar app on the same date and month, but in the year 2090. To avoid issues related to time-sensitive information or changing event details, we decoupled the evaluation criteria from the actual event date, ensuring that the correctness of task completion is independent of the assessment time.

This systematic approach to task collection and design enables comprehensive evaluation of an agent's ability to reason globally, manage inter-app dependencies, and execute complex workflows that mirror real-world user demands in multi-application environments. In the lower part of Figure 3, we provide examples to illustrate how collaborative tasks involving two applications can be constructed.

## A.5 DETAILS OF EVALUATION METRICS

### A.5.1 EVALUATION METRICS FOR L1-GUI CONETNT UNDERSTANDING.

Formally, the accuracy for an evaluation set comprising $N$ Question-Options-Answer pairs can be defined as:

$$\text{Acc} = \frac{1}{N} \sum_{i=1}^{N} \Theta(o_i^* = o_i),$$

(5)

where $\Theta(o_i^* = o_i)$ is an indicator function that equals 1 if the predicted answer $o_i^*$ for the $i$-th pair matches the ground-truth answer $o_i$ and 0 otherwise.

To account for variations in the number of answer choices, we introduce a simple dynamic adjustment factor $\alpha$ to rescale the original accuracy of each question. Taking Windows platform which has $N_{win}$ questions as an example, the accuracy of L1 is computed as:

$$\text{Acc}_{win} = \frac{1}{N_{win}} \sum_{i=1}^{N_{win}} \alpha \cdot \Theta(o_i^* = o_i), \quad \alpha = \frac{m_i - 1}{m_i}$$

(6)

where $m_i$ is the number of options for question $i$. Accordingly, for any given difficulty level, the agent's understanding ability (i.e., accuracy) can be computed as:

$$\text{Score} = \sum_{j \in \mathcal{O}} \frac{N_j}{N} \cdot \text{Acc}_j$$

(7)

where $\mathcal{O} = \{\texttt{win}, \texttt{linux}, \texttt{mac}, \texttt{ios}, \texttt{android}, \texttt{web}\}$ denotes the set of operation platforms, $N_j$ is the number of questions for platform $j$, $N = \sum_{j \in \mathcal{O}} N_j$ is the total number of questions across all platforms.

### A.5.2 EVALUATION METRICS FOR L3 AND L4

We define EQA as a continuous-time recall metric over cumulative agent effort. Consider an ordered set of $N$ tasks. For each task $i \in \{1, 2, \ldots, N\}$, let:

- $s_i = 1$ if the agent successfully completes task $i$, and $s_i = 0$ otherwise,
- $t_i > 0$ be the number of steps the agent takes to complete task $i$.

We define the cumulative cost and cumulative success after the first $k$ tasks as:

$$T_k = \sum_{j=1}^{k} t_j, \qquad S_k = \sum_{j=1}^{k} s_j.$$

(8)

Let the global budget be $T_{\max} = N \cdot t_{\max}$, where $t_{\max}$ is the maximum step limit per task. We normalize the cumulative effort as:

$$u_k = \frac{T_k}{T_{\max}} \in [0, 1].$$

(9)

The instantaneous recall at normalized time $u$ is defined as:

$$R(u) = \max_{k:\, u_k \leq u} \frac{S_k}{N}, \qquad u \in [0, 1].$$ (10)

Finally, EQA is computed as the area under the step-wise non-decreasing recall curve:

$$\text{EQA} = \int_0^1 R(u)\, du \;\approx\; \frac{1}{M} \sum_{m=0}^{M-1} R\left(\frac{m}{M-1}\right),$$ (11)

where $M = 101$ denotes the number of uniformly spaced evaluation points. This metric encourages agents to complete more tasks in fewer steps, offering a holistic measure of task performance.

### A.6 Further Analysis and Key Findings

In this section, we conduct an in-depth analysis to delve into the underlying causes and implications reflected in our benchmark results. Our investigation is structured around three primary dimensions: platform, task, and model, and adheres to a single-variable control principle to ensure the validity of our comparisons. By processing the empirical results, we identify the essential challenges facing contemporary GUI agents, offering valuable guidance to advance future development.

**Finding 1: General-purpose language models excel at task decomposition, planning, and self-reflection but struggle with fine-grained visual interactions.** Across different model categories, proprietary models, exemplified by GPT-4o and Claude, demonstrate pronounced limitations in fine-grained GUI tasks. As shown in Table 2 and the right part of Figure 4, their average scores in L2 are merely 2.87 for GPT-4o and 4.66 for Claude-3.7, in contrast to the specialized visual grounding model UGround-V1-7B, which achieves a score of 65.68%. A similar trend emerges in L3 tasks. For instance, GPT-4o alone achieves success rates (SR) of only 4.05%/6.13% in single-app automation scenarios (Max Step = 15/50, see Table 3). However, when paired with domain-specific grounding modules such as UGround-V1-7B or UI-TARS-1.5-7B, the SR of GPT-4o rises substantially to 11.93%/17.50%. These phenomena indicate that proprietary models are inherently deficient in the precise perception and localization of UI components, which can be effectively remedied by specialized perception modules.

Thus, beyond incorporating auxiliary localization modules during training and increasing the amount of fine-grained perceptual data, a more fundamental and forward-looking direction lies in embracing a modular architecture. This approach enables the model to dynamically interface with external modules based on its own capability gaps (e.g., visual grounding), effectively allowing for targeted augmentation through specialized "external agents". Within this synergistic and collaborative framework, the capabilities of general-purpose models can be significantly augmented and tailored for complex GUI automation tasks.

**Finding 2: Accurate visual grounding significantly determines the success rate of GUI task execution.** The full decision-making pipeline of a GUI agent can be abstracted into three stages: perceive accurately $\Rightarrow$ reason properly $\Rightarrow$ act precisely. An initial failure in element localization will cause cascading errors, rendering subsequent steps ineffective. To examine the critical role of localization, we designed two complementary experimental setups as shown in the left part of Figure 4: (1) fixing the planner while incrementally improving the grounder, and (2) fixing the grounder while varying the planner. Correlation analyses revealed a clear pattern: with the same planner, improving localization alone led to a $2.8\times$ ($\Delta = 17.25$) increase in SR. In contrast, when localization performance remained roughly constant, replacing the planner with a stronger VLM yielded marginal returns ($1.15\times$, $\Delta = 3.58$). This finding indicates that the key to improving GUI task automation lies in advancing visual localization. Consequently, the visual grounder should be the primary and most critical component in any modular architecture, providing the stable foundation required for higher-level functions like planning, memory, and reflection.

**Finding 3: Efficiency, including step minimization and early stopping, is a critical yet underexplored dimension of GUI agent performance.**

The introduction of the EQA metric enables us to move beyond evaluating whether an agent simply completes a task, by shifting attention to how efficiently the task is accomplished. This novel perspective facilitates deeper insights through a more fine-grained analysis of agent behavior.

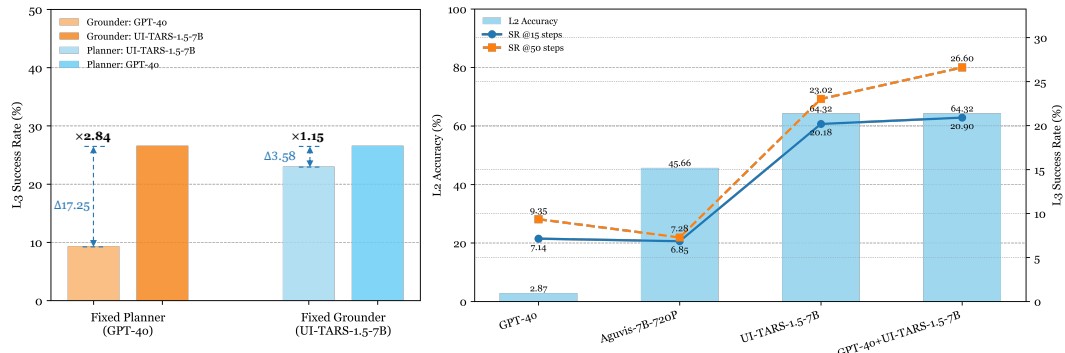

Figure 4: **Left:** Demonstrates the relative contribution of visual grounding versus planning in driving performance gains under current conditions. We consider two experimental conditions—fixing the planner while varying the grounder, and vice versa—and examine how different combinations affect task success rate. Similar color hues denote groups with the same fixed planner or grounder. **Right:** Task success grows roughly linearly with visual-grounding accuracy. General-purpose language models are virtually "blind" at the L2 grounding stage, which drives their L3 automation success rate (SR) sharply down. Plugging in a dedicated visual grounder restores precise perception and, in turn, lifts SR dramatically—highlighting fine-grained grounding as the principal bottleneck.

We additionally compute two derived metrics, $\frac{EQA}{SR}$ and $SR - EQA$, to facilitate a more comprehensive analysis. Based on the definition of the EQA and SR, we further reformulate them as:

$$EQA = \frac{1}{N} \sum_{i \in \mathcal{C}} (1 - u_i), \;\; SR = \frac{|\mathcal{C}|}{N}, \tag{12}$$

where $\mathcal{C}$ denotes the set of all successfully completed tasks, and $u_i = \frac{T_i}{T_{max}} \in (0, 1]$ represents the normalized completion step of task $i$ within the global step budget. From this, $\frac{EQA}{SR}$ and $SR - EQA$ can be derived as:

$$\frac{EQA}{SR} = \frac{1}{|\mathcal{C}|} \sum_{i \in \mathcal{C}} (1 - u_i) = 1 - \frac{1}{|\mathcal{C}|} \sum_{i \in \mathcal{C}} u_i, \tag{13}$$

$$SR - EQA = \frac{1}{N} \sum_{i \in \mathcal{C}} u_i = SR \times (\frac{1}{|\mathcal{C}|} \sum_{i \in \mathcal{C}} u_i), \tag{14}$$

where $\frac{1}{|\mathcal{C}|} \sum_{i \in \mathcal{C}} u_i$ denotes the average steps in which a task is completed.

Therefore, $\frac{EQA}{SR}$ has an intuitive physical interpretation: it reflects the average remaining steps per successful task. Its upper bound is 1, which corresponds to the idealized case where all successful tasks are completed almost immediately (i.e., at the first step). Conversely, its lower bound is 0, indicating that all successful completions occur only at the very end of the allowed budget. $\frac{EQA}{SR}$ quantifies how many steps, on average, are consumed before successful completion. Meanwhile, $SR - EQA$ also has an intuitive physical interpretation: it is approximately proportional to the total normalized time consumed across all successful tasks, and can be interpreted as a "redundant step bill". A larger difference between EQA and SR implies a greater average normalized completion time $u_i$ for the successful set, meaning that tasks tend to be completed closer to the end of the budget—i.e., with more redundant steps. Conversely, a smaller difference (approaching zero) indicates that most successful tasks are completed early, near the beginning of the budget, suggesting minimal or no redundancy. Thus, the magnitude of the gap between EQA and SR effectively captures how "wasteful" the agent is, even among the tasks it completes.

We re-organize the $\frac{EQA}{SR}$ and $SR - EQA$ using the average results in Table 3 as EQ[1] and EQ[2], and present the aggregated findings in Table 5. Combining with Figure 5, we can disclose four complementary patterns. First, the modular pairing of a powerful planner with a specialized grounder, exemplified by GPT-4o + UGround-V1-7B and GPT-4o + UI-TARS-1.5-7B, elevates the success rate under a 50-step budget by roughly 5.7%, yet still incurs a substantial redundant step cost (EQ[2] = 7-8),

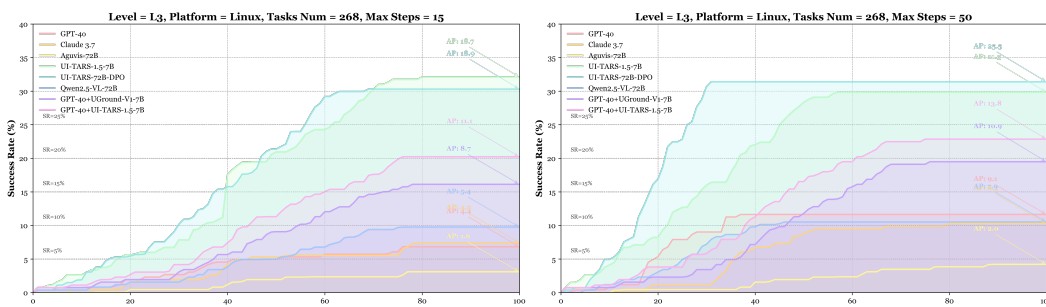

Figure 5: **EQA visualization across different models under L3 for different allowed steps.** As discussed in Section 2.4, EQA reflects a combination of task completion and efficiency (i.e., the number of steps used upon completion). In practice, we compute it by interpolating both the step budget and the success rate (SR) 100 times. The area under the curve formed by these interpolated SR values yields the final EQA score.

Table 5: Additional metrics derived by SR and EQA. Here, $EQ_{15}^1$ and $EQ_{15}^2$ denotes for $\frac{EQA}{SR}$ and $SR - EQA$, respectively, when the maximal step is 15. $\Delta EQ^1 = EQ_{50}^1 - EQ_{15}^1$ and so is the $\Delta EQ^2$. Similarly, $\Delta SR = SR_{50} - SR_{15}$ and $\Delta EQA = EQA_{50} - EQA_{15}$

| Model | $\Delta SR$ | $\Delta EQA$ | $EQ_{15}^1$ | $EQ_{50}^1$ | $EQ_{15}^2$ | $EQ_{50}^2$ | $\Delta EQ^1$ | $\Delta EQ^2$ |
|---|---|---|---|---|---|---|---|---|
| GPT-4o (2024) | 2.21 | 2.08 | 0.57 | 0.66 | 3.09 | 3.22 | 0.09 | 0.13 |
| Claude-3.7 (2025) | 1.20 | 0.95 | 0.50 | 0.55 | 3.40 | 3.65 | 0.04 | 0.25 |
| Aguvis-72B (2024b) | 0.43 | 0.56 | 0.52 | 0.57 | 3.29 | 3.16 | 0.05 | -0.13 |
| UI-TARS-1.5-7B (2025) | 2.84 | 3.23 | 0.64 | 0.70 | 7.31 | 6.92 | 0.06 | -0.39 |
| UI-TARS-72B-DPO (2025) | 2.06 | 5.27 | 0.62 | 0.77 | 8.96 | 5.75 | 0.16 | -3.21 |
| Qwen2.5-VL-72B (2025) | 1.57 | 2.86 | 0.60 | 0.74 | 4.91 | 3.62 | 0.14 | -1.29 |
| GPT-4o+UGround-V1-7B (2024) | 5.71 | 5.57 | 0.62 | 0.70 | 7.43 | 7.57 | 0.08 | 0.14 |
| GPT-4o+UI-TARS-1.5-7B (2025) | 5.70 | 7.53 | 0.53 | 0.70 | 9.74 | 7.91 | 0.17 | -1.83 |
| Avg. | 2.72 | 3.51 | 0.57 | 0.67 | 6.02 | 5.23 | 0.10 | -0.79 |

signaling that cross-module coordination and early termination heuristics remain inadequate. Second, the large-scale DPO-aligned UI-TARS-72B-DPO achieves the strongest efficiency profile, increasing $EQ^1$ to 0.773 while compressing $EQ^2$ from 8.96 to 5.75 ($\Delta EQ^2 = -3.21$); this demonstrates that aligning to human preferences that explicitly reward rapid task completion can translate directly into tangible efficiency gains. Third, general-purpose agents such as GPT-4o and Claude-3.7 extract minimal benefit from a longer budget ($\Delta SR < 2.5\%$) and even exhibit higher redundant step costs ($EQ^2$ increases from 3.09 to 3.22 and 3.40 to 3.65, respectively), underscoring that simply extending the interaction horizon cannot compensate for their limited visual granularity and action precision, therefore, integrating specialized perception or actuation modules is becoming indispensable. Lastly, none of the curves in Figure 5 attains the ideal "hug-the-top-left-corner" profile, underscoring a pervasive lack of effective early-stopping heuristics and cost-aware search strategies.

To mitigate the efficiency bottlenecks aforementioned, we identify three possible research avenues. (1) Confidence- or value-based early-termination policies: equip agents with stopping rules that immediately end an episode when the marginal utility of further actions falls below a threshold, rather than passively consuming the entire step budget. (2) Cost-sensitive fine-tuning: during reinforcement-learning (or DPO-style) alignment, impose explicit penalties for every superfluous action so that optimization shifts from maximizing success rate (SR) alone to jointly maximizing the success-conditioned efficiency score EQA. (3) Progress-aware self-reflection: require the planner to periodically estimate the set of remaining sub-goals and, upon detecting that all objectives are satisfied, issue an immediate FINISH action. Together, these interventions target the twin goals of cutting redundant steps and encouraging agents to "know when to stop", thereby narrowing the gap between current GUI agents and human-level operational efficiency.

**Finding 4: The limitation of action space restricts the agent's ability to execute planned actions, especially in GUI task collaboration scenarios.**

In Table 3, a notable fraction of models fail to complete the task on the web platform. The underlying cause is that, during web-interaction execution, the models lack the ability to trigger action `switch_tab` to enable 'press Tab to switch tabs'. In headless-browser settings, this omission blocks seamless navigation across multiple tabs, preventing cross-window information from being transferred from one context to another and ultimately derailing task completion.

On the other hand, due to the inherent heterogeneity of interactions across desktop, mobile, and web platforms, the current prompt-based definition of action functions struggles to comprehensively capture the full spectrum of platform-specific operations. Moreover, during inference, models may confuse actions across platforms, producing incorrect or incompatible output actions. Such issues can directly lead to task failure, even in single-platform, multi-app scenarios, and become particularly pronounced in multi-platform, multi-app settings, for example, when copying text from a web page and pasting it into a desktop application like Word for further formatting.

Building on these observations, we argue that a more generalizable, extensible, and potentially platform-agnostic definition of the action space is worth pursuing. One intuitive and straightforward direction is to construct a unified API abstraction layer that comprehensively covers multi-platform operations. Under this design, the agent interacts with the environment by invoking platform-independent APIs, while the backend of the API is responsible for platform-specific adaptations. An alternative route focuses on operation atomization. Unlike current action spaces that rely on fixed, platform-tied commands, an ideal action space would emphasize a set of primitive operations, decoupled from any particular environment. Agent-issued instructions are then mapped to these primitives via a many-to-many translation schema, where each high-level intent may correspond to a combination of atomic steps. These atomic units can then be recompiled into platform-specific execution commands, enabling robust and consistent interaction across environments. Beyond these two approaches, we believe that the research community should continue to explore better formulations of the action space, those characterized by strong generality, high extensibility, and minimal platform dependence.

**Finding 5: Although many GUI agents excel in simple cases, their effectiveness diminishes significantly as task complexity rises, revealing limited generalization capabilities.**

As shown in Figure 6, although many systems perform impressively on easy scenarios, their accuracy/success rate deteriorates sharply as soon as either (i) the local difficulty within a level increases (easy → medium → hard; basic → advanced) or (ii) the global task complexity rises from L1 to L4. These steep drops - especially pronounced for general-purpose LLMs - indicate that today's agents still lack robust generalization to harder, less stereotyped GUI situations. For example, the GUI understanding score of GPT-4o drops from 60.2% (easy) to 53.5% (hard), a -11% decrease, while even the highly tuned InternVL3-72B loses 4% (Table 1). In element grounding, switching from 'Basic' to semantically implicit 'Advanced' queries slashes GPT-4o's mean accuracy by nearly 40% and still costs the specialist UI-TARS-72B-DPO 16% (Table 2). The effect compounds across levels: the strongest agent (GPT-4o + UI-TARS-1.5-7B) succeeds in 26.6% of tasks at L3 but only 8.8% once multi-app collaboration is required in L4, a 67% collapse that is mirrored by other models (Tables 3). Concomitant declines in EQA confirm that agents not only fail more often but also waste proportionally more steps before failing.

These sharp drops expose three intertwined bottlenecks: (1) ill-posed perceptual clues (small widgets, non-salient text), (2) longer credit-assignment chains, and (3) noisy action spaces inflate the search space exponentially. Current models, trained largely on static screenshots, lack the robust abstract representations and error-driven exploration strategies needed to cope.

Possible targeted remedies include: (1) Curriculum & hard-negative mining. Intentionally up-sample adversarial layouts (occlusion, theme changes, deceptive affordances) during instruction tuning to inoculate perceptual modules against distribution shift. (2) Dynamic skill routing. Teach planners to self-diagnose uncertainty and automatically invoke auxiliary skills (OCR, vision transformers, memory retrieval) as difficulty rises. (3) Hierarchical planners with macro-actions. Introduce option-level abstractions (e.g., `open-browser-tab`) so that sparse EQA-style rewards can flow to high-level decisions instead of individual clicks. (4) Unified state schema for all applications. Store "App → Page → Element" graphs in an external memory that survives context switches, allowing the planner to reason over shared entities rather than raw pixel buffers.

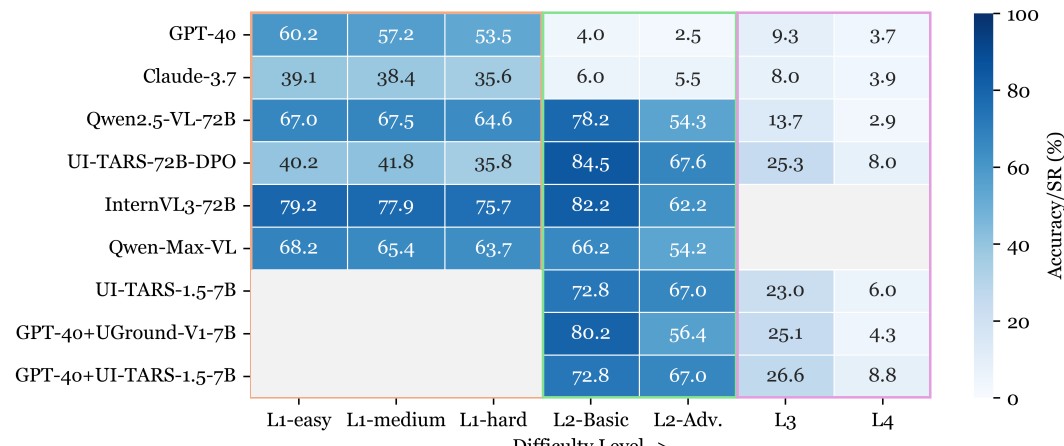

Figure 6: **Difficulty-Gradient Heatmap.** Models' scores across difficulty levels are encoded with a single-hue palette whose saturation fades from high (dark) to low (light). Colored rectangles outline comparable model groups. Within and across these groups, the color consistently fades from L1, L2 to L3 and L4, indicating that higher task complexity amplifies each model's weaknesses and causes a steep performance drop-off.

We believe that by attacking these verified failure modes, the community can turn today's hardest cases, from implicitly described buttons to multi-window workflows, into stepping-stones toward truly general-purpose GUI agents.

**Finding 6: The failures in multi-application environments primarily stem from limited cross-context memory and action space, rather than issues with perception or planning.**

Success drops that cannot be explained by harder screenshots or longer action chains alone appear as soon as the agent must pass information between applications. The strongest single-app system, GPT-4o+UI-TARS-1.5-7B, falls from **26.6%** SR on L3 to just **8.8%** on L4 (Tables 3); UI-TARS-72B-DPO shows an almost identical collapse (25.3% to 8.0%). Failures concentrate at window or tab boundaries: five models are labeled '-' on the Web platform simply because they cannot express the primitive switch_tab. At the same time, EQA shrinks far more than the accompanying $SR - EQA$ penalty (e.g., 18.7% $\rightarrow$ 6.4% for GPT-4o + UI-TARS), signaling that agents waste many steps rediscovering the context they have just lost. These phenomena point to a deficit in working memory and action-space coverage, rather than in perception or generic planning.

Addressing these failures may require agents to focus on memory-centric research avenues, including: (1) External episodic buffer. Log every UI observation and write-back (*copy*, *navigate*, *paste* ...) to an append-only timeline that the language planner can query with natural language—much like retrieval-augmented generation, but for GUI states. (2) Semantic anchors. Tag entities (e.g., "flight-price $514") with stable IDs when first seen; subsequent references use the anchor, so the planner no longer depends on window focus to recall an object. (3) Cross-context consistency checks. Inject lightweight assertions, for example, "clipboard should now contain X" and "target window title equals Y". Violations trigger immediate self-repair instead of long, fruitless trial-and-error loops, cutting the redundant steps that dominate L4 failures.

## A.7 BENCHMARKING DETAILS

To ensure fairness, we evaluated all candidate models through a unified interface compatible with the OpenAI API protocol. Specifically, each model was deployed as an API-style service, and outputs were obtained by sending POST requests to the service endpoint along with the conversation input. For each model, we crafted both system and user prompts strictly based on official documentation or released code. For proprietary models, we designed detailed and effective prompts to elicit high-quality responses as faithfully as possible. Apart from model-specific settings, all other parameters, such as temperature and top-p, were kept consistent across evaluations.

During evaluation, the input and output processing pipeline was tailored to the requirements of each task level. For L1-GUI Content Understanding and L2-GUI Element Grounding, the input to the model comprised the GUI screenshot paired with either the relevant instruction or the question-options set. Model outputs were assessed using `exact-match` evaluation protocol, analogous to standard practices in grounding and QA tasks. However, given the variability in instruction-following abilities across different models, for example, the QA tasks in L1, we observed that some model outputs could not be reliably parsed. To address this, we implemented a hybrid parsing mechanism based on multiple regular expressions to robustly extract valid answers. In our codebase, we expose a customizable `parse_function` for each method, enabling tailored post-processing strategies to accommodate the unique output formats of various models.

For L3-GUI Task Automation and L4-GUI Task Collaboration, evaluation focused solely on whether the agent successfully achieved the desired end state, without the need to interpret intermediate natural language outputs. Therefore, parsing functions were not required for these levels; instead, we compared the final state directly against predefined success criteria to determine task completion.

