# OpenReview forum: "MMBench-GUI: Hierarchical Multi-Platform Evaluation Framework for GUI Agents"
_ICLR.cc/2026/Conference — ICLR 2026 Conference Withdrawn Submission_

### Official Review · Reviewer_ri1F · 2025-10-27

**Soundness:** 3
**Presentation:** 3
**Contribution:** 3
**Rating:** 6
**Confidence:** 3

**Summary:**

This paper presents a new benchmark for CUA agents, which supports multiple platforms. The work is technically solid and well-done, providing a structured way to assess agents across perception, grounding, automation, and multi-application collaboration tasks. The tests of various schemes on this benchmark also sound reasonable, providing essential information for researchers in this field to further explore.

**Strengths:**

This is one of the first to unify GUI agent evaluation across platforms and hierarchical capability levels, through low-level perception (understanding, grounding) to high-level reasoning (automation, collaboration). I like the four-level evaluation structure.

The proposed EQA metric also sounds good, which could address an issue ignored by prior success-only metrics.

The experiments on this benchmark by including major open-source and proprietary VLM/LLM systems are also well done, which is beneficial to the research on CUA agents. And also, it's good to see the results show that accurate visual grounding is the key bottleneck.

**Weaknesses:**

The paper’s primary contribution is the benchmark design, not new algorithmic or modeling techniques. It's useful to the CUA research, but it's hard to say if its technical contributions are sufficient as an ICLR paper.

And also, prior works like OSWorld already explore parts of these capabilities, and the paper’s novelty lies mainly in unifying them, not in entirely new data or task types.

**Questions:**

In addition to the novelty issue, the authors are also suggested to add more systematic discussions on the models/solutions' generalization capabilities evaluated with this benchmark, e.g., performance on unseen apps.

---

### Official Review · Reviewer_oqiV · 2025-10-29

**Soundness:** 2
**Presentation:** 2
**Contribution:** 2
**Rating:** 2
**Confidence:** 5

**Summary:**

This paper focuses on evaluating GUI automation agents through a hierarchical benchmark, MMBench-GUI, which spans six platforms and four capability levels, introducing an Efficiency–Quality-Aware (EQA) metric to measure task success and efficiency across more than 8,000 tasks. It does not propose any new model, algorithm, or representation-learning insight.

**Strengths:**

1. Benchmark covering six major platforms and four hierarchical capability levels.
2. Introduces an Efficiency–Quality-Aware (EQA) metric combining accuracy and efficiency.
3. Offers extensive empirical results across a wide range of proprietary and open-source GUI agents.
4. Proposes a transparent and well-documented data construction process.

**Weaknesses:**

1. **No contribution to learning representations or algorithms.**
   The work is purely an evaluation benchmark with no proposed model, training method, or learning insight, misaligned with ICLR’s core focus on representation learning.

2. **Limited conceptual novelty.**
   The four-level hierarchy (understanding → grounding → automation → collaboration) closely mirrors existing works such as OSWorld, ScreenSpot-Pro, and UI-TARS, amounting to a reorganization rather than a conceptual advance.

3. **Lack of benchmark reliability evidence.**
   The paper does not report human baselines, inter-annotator agreement, or quality-control statistics, making it unclear whether the dataset accurately reflects human task understanding.

4. **No analysis of learned representations.**
   The paper reports only performance metrics and provides no investigation into internal representations, generalization behavior, or failure structures of the evaluated models.

5. **Evaluation is descriptive rather than explanatory.**
   While the results show performance differences, there is no deeper causal or diagnostic analysis that connects outcomes to representational or architectural factors.

8. **Primarily an engineering contribution.**
   The main value lies in data collection and system integration rather than new scientific understanding or learning principles.

9. **Efficiency metric (EQA) adds limited theoretical value.**
   Although formally defined, EQA is a straightforward success-vs-step measure and offers minimal conceptual advancement beyond existing efficiency metrics.

**Questions:**

1. **Interpretation of EQA metric.**
   The EQA metric measures combined efficiency and accuracy, but its practical meaning is unclear. Could the authors provide examples illustrating how EQA changes agent ranking compared to success rate alone, and whether this correlates with human judgment of efficiency?
2. **Cross-application task design.**
   For L4 “Task Collaboration,” can the authors describe how multi-application coordination is simulated? Are tasks executed in real interactive environments, or in scripted virtual contexts?
3. **Scalability and potential for training use.**
   Is it feasible to scale MMBench-GUI beyond the current size, particularly for the higher-level (L3 and L4) tasks? Additionally, could the benchmark be adapted or extended for *training* purposes—for example, to support representation learning or reinforcement learning of GUI agents—rather than evaluation only? If so, what learning objectives or architectures would benefit from it?
4. **Human performance reference.**
   Did the authors measure human accuracy or efficiency on any subset of tasks? Such baselines would help calibrate the benchmark’s difficulty and interpret model performance.

---

### Official Review · Reviewer_ddvL · 2025-10-30

**Soundness:** 3
**Presentation:** 3
**Contribution:** 3
**Rating:** 6
**Confidence:** 4

**Summary:**

This paper introduces MMBench-GUI, a well-designed hierarchical benchmark that evaluates GUI agents across four levels and six platforms, with 8000+ tasks, and a novel and valuable EQA efficiency metric. The comprehensive evaluation framework provides useful insights for the community. The extensive experiments across 10+ models yield actionable findings about visual grounding bottlenecks and efficiency gaps. While the work relies on existing benchmarks and LLM-generated data, and some findings are expected, the systematic unification, broad platform coverage, and practical contributions make this a useful benchmark paper that will benefit GUI agent research.

**Strengths:**

S1: The four-level framework naturally decomposes GUI agent capabilities from basic understanding to complex collaboration, enabling fine-grained diagnosis.

S2: The benchmark spans all major platforms (Windows, macOS, Linux, iOS, Android, Web) and has a balanced data distribution, addressing a gap in existing work.

S3: Testing 10+ models with detailed analysis across platforms and difficulty levels provides useful insights for the community.

**Weaknesses:**

W1: L1/L2 rely heavily on automated generation via Claude/GPT models, with limited details on quality control using manual sampling. For manual sampling, what percentage of LLM-generated questions were rejected during manual review? What specific issues were found? More details could be provided.

W2: The continuous-time integral formulation of EQA (Equation (11)) is somewhat opaque. Have the authors compared EQA to other efficiency metrics? How sensitive is it to the choice of M=101 evaluation points? Could the authors show that EQA correlates with human judgments of efficiency?

**Questions:**

Q1: How were the 15 and 50 step limits determined? What happens with other budgets?

Q2: For L3/L4, what percentage of tasks are newly created vs. curated from existing benchmarks? Can the authors clarify the contribution breakdown?

---

### Official Review · Reviewer_hau5 · 2025-11-07

**Soundness:** 3
**Presentation:** 1
**Contribution:** 2
**Rating:** 2
**Confidence:** 4

**Summary:**

The paper introduces a benchmark to evaluate the various capabilities required in GUI agents. The task is built in a hierarchical manner, from understanding to perception, to two stages of action prediction and planning. The first stage (L3) deals with single-platform tasks, and the second stage (L4) deals with multi-platform tasks. The authors evaluate several GUI agents on their benchmarks and through analysis, the authors point out several areas where GUI agents underperform.

**Strengths:**

1. The papers exhaustively cover different abilities of GUI agents and diverse and different platforms. The motivation to do this hierarchically is sound.
2. The authors introduce MacOSArena, which includes 70 tasks for MacOS, is a novel contribution. This is in addition to L1 and L2 images they curate and the tasks they create for the same.
3. The paper analyzes the efficiency of GUI agents in performing tasks in addition to task success. Such a type of evaluation is much needed.

**Weaknesses:**

1. My main point of concern is that this benchmark does not give new insights regarding GUI agents. I do agree that unifying different tasks in one framework is useful, but I don't see how doing so allows authors to come up with new insights (please see my comments below to know my problems with the tasks). Overall, I feel several works already point out the issues that the authors learn from this benchmark, which brings an important question as to its practical utility.
	1. For L1, a related work is GUI-World [1] already covers a plethora of platforms and several software applications, and tests both image and video models on a number of prediction, reasoning, and captioning tasks.
	2. For L2, ScreenSpotv2, ScreenSpot-Pro [3] and UI-Vision [4] already have combined over 100 different software applications covered. Moreover, for the authors' claims on direct instructions L195, Ui-vision has functional and spatial instructions that go beyond direct instructions.
	3. For L3 and L4, the tasks are already in the existing benchmark, and some works analyse models' weakness across these as they are very popular benchmarks (except MacOS, which I already pointed to in the strengths)
	There might be other relevant works that I missed. Moreover, many of the insights the paper provides are also covered in related works. All this makes me wonder about the need for a new benchmark (see problems listed below for more details) especially given that most of L3 and L4 are from already existing tasks. If the authors could provide a clear and convincing argument in this regard, it would be really helpful.
	**Note:** I don't mean to say this negatively. I know that the authors have put a lot of effort into this work, and I appreciate and respect that. But from a conference point of view, I believe these questions need to be addressed for a benchmark.
2. Where do the authors get their images from? Currently, L921-928 only tells us that the screenshots are obtained manually from diverse software applications, but neither the coverage of these applications nor their sources is described. This hinders the transparency. Did the authors pick the images themselves? If so, what criteria were used, and how did the authors ensure that there was no bias?
3. **Some concerns around the L1-L4:**
	1. For L1 and L2 tasks, there is too much reliance on LLMs to create the tasks. This risks biases of the LLMs into the annotation. For example, for the L1 tasks, the LLM might ask questions around elements that it is confident about, which might cause biases in the initial distribution of questions/instructions.
	2. The performance of models on L1-L2 is quite high. For L2, the basic category, which I believe to be the most important category, as in realistic cases, agents would receive direct information from a planner or human, the performance is quite high. The authors have not evaluated more recent models like [5], [6], which further raises the question of how high the accuracy might go.
	3. How much manual labelling was done for the L2 task? And if a limited number of UI elements were annotated, what motivated this choice?
	4. For the MacOSArena, could the authors give more details regarding how the tasks were created?
4. Most of the meat of the paper is in the appendix. I understand that the authors have a lot of content to present, but even the main contribution, which is the EQA metric, is in the appendix. This makes the read a little tiring as one needs to go to the appendix to understand all the details (including the source of images). I believe the details regarding the source and metrics are very important for a benchmark paper and need to be summarised appropriately in the main text. One suggestion here is to shorten the task definitions as these are not novel and already known, and let the new readers refer to the appendix for more details on the task definition.
5. There is no detailed comparison with related benchmarks. A good table highlighting what new additions the authors present could be helpful.

[1] Chen et al. GUI-World: A Dataset for GUI-Oriented Multimodal Large Language Models

[2] Cheng et al. SeeClick: Harnessing GUI Grounding for Advanced Visual GUI Agents

[3] Li et al. ScreenSpot-Pro: GUI Grounding for Professional High-Resolution Computer Use

[4] Nayak et al. UI-Vision: A Desktop-centric GUI Benchmark for Visual Perception and Interaction

**Questions:**

Please refer to the weakness for more questions. The question below is a clarification I wanted.
1. I had a question around the EQA metric. From my understanding, this metric is sensitive to the order of the tasks performed, as the order determines T_k and S_k. Am I right here? If so, then how is this a reliable metric, especially since the tasks are ordered randomly and the agents can't choose the order? Do all the people who evaluate the tasks have to run them in a particular order?

---

### Note · Authors · 2025-11-13

**Comment:**

Dear AC, reviewers,

Thank you very much for your hard work and careful review. We have carefully gone through your comments and agree that the current writing and chapter organization indeed need further refinement. Some details regarding the data construction also need to be described in greater depth. Your review has been very inspiring for us, and we would like to express our sincere gratitude for your insightful feedback. After considering your comments and the whole timeline of rebuttal period, we decide to withdraw our paper to make further polishment.

Best regards,
Authors

**Withdrawal Confirmation:**

I have read and agree with the venue's withdrawal policy on behalf of myself and my co-authors.